

**Estimation of Reactive Inorganic Iodine Fluxes in the Indian and Southern Ocean Marine**
**Boundary Layer**
Swaleha Inamdar[1,2], Liselotte Tinel[3], Rosie Chance[3], Lucy J. Carpenter[3], Prabhakaran Sabu[4],
Racheal Chacko[4], Sarat C. Tripathy[4], Anvita U. Kerkar[4], Alok K. Sinha[4], Parli Venkateswaran
Bhaskar[4], Amit Sarkar[4,5], Rajdeep Roy[6], Tomas Sherwen[3,7], Carlos Cuevas[8], Alfonso Saiz-
Lopez[8], Kirpa Ram[2] and Anoop S. Mahajan[1]*
[1]Centre for Climate Change Research, Indian Institute of Tropical Meteorology, Dr Homi
Bhabha Road, Pashan, Pune, 411 008, India
[2]Institute of Environment and Sustainable Development, Banaras Hindu University, Varanasi,
005, India
[3]Wolfson Atmospheric Chemistry Laboratories, Department of Chemistry, University of York,
YO10 5DD, UK
[4]National Centre for Polar and Ocean Research, Goa, 403 804, India
[5]Environment and Life Sciences Research Centre, Kuwait Institute for Scientific Research
Centre, Al-Jaheth Street, Shuwaikh, 13109, Kuwait
[6]National Remote Sensing Centre, Department of Space Government of India Balanagar,
Hyderabad, 500 037, India
[7]National Centre for Atmospheric Science, University of York, York YO10 5DD, UK
[8]Department of Atmospheric Chemistry and Climate, Institute of Physical Chemistry
Rocasolano, CSIC, Madrid, Spain.
* Corresponding author: Anoop S. Mahajan (anoop@tropmet.res.in); phone: +91 20 2590 4526



## Abstract

Iodine chemistry has noteworthy impacts on the oxidising capacity of the marine boundary layer (MBL) through the depletion of ozone ($O_3$) and changes to $HO_x$ (OH/$HO_2$) and $NO_x$ (NO/$NO_2$) ratios. Hitherto, studies have shown that the reaction of atmospheric $O_3$ with surface seawater iodide ($I^-$) contributes to the flux of iodine species into the MBL mainly as hypoiodous acid (HOI) and molecular iodine ($I_2$). Here, we present the first concomitant observations of iodine oxide (IO), $O_3$ in the gas phase, and sea surface iodide concentrations. The results from three field campaigns in the Indian Ocean and the Southern Ocean during 2014-2017 are used to compute reactive iodine fluxes to the MBL. Observations of atmospheric IO by MAX-DOAS show active iodine chemistry in this environment, with IO values up to 1 pptv (parts per trillion by volume) below latitudes of 40°S. In order to compute the sea-to-air iodine flux supporting this chemistry, we compare previously established global sea surface iodide parameterisations with new, region-specific parameterisations based on the new iodide observations. This study shows that regional changes in salinity and sea surface temperature play a role in surface seawater iodide estimation. Sea-air fluxes of HOI and $I_2$, calculated from the atmospheric ozone and seawater iodide concentrations (observed and predicted), failed to adequately explain the detected IO in this region. This discrepancy highlights the need to measure direct fluxes of inorganic and organic iodine species in the marine environment. Amongst other potential drivers of reactive iodine chemistry investigated, chlorophyll-*a* showed a significant correlation with atmospheric IO (R = 0.7 above the 99 % significance level) to the north of the polar front. This correlation might be indicative of a biogenic control on iodine sources in this region.

Keywords: iodine, Southern Ocean, Indian Ocean, marine boundary layer



## 1. Introduction

Iodine chemistry in the troposphere has gained interest over the last four decades after it was
first discovered to cause depletion of tropospheric ozone ($O_3$) (Chameides and Davis, 1980;
Jenkin et al., 1985) and cause changes to the atmospheric oxidation capacity (Davis et al., 1996;
Read et al., 2008). Iodine studies in the remote open ocean are important considering its role
in tropospheric ozone destruction (Allan et al., 2000), the formation of potential cloud
condensation nuclei and impact on cloud radiative properties (McFiggans, 2005; O'Dowd et
al., 2002). However, iodine chemistry in the remote open ocean is still not completely
understood, with uncertainties remaining around the sources and impacts of atmospheric iodine
(Saiz-Lopez et al., 2012; Simpson et al., 2015).
Recent studies of atmospheric iodine chemistry have focused on the detection of iodine oxide
(IO) in the marine boundary layer (MBL) as a fingerprint for active iodine chemistry. IO may
itself also participate in particle nucleation if present at high concentrations (Saiz-Lopez et al.,
2006b). Iodine containing precursor compounds undergo photo dissociation to produce iodine
atoms (I), which rapidly react with ambient ozone, forming IO (Chameides and Davis, 1980).
Until recently, fluxes of volatile organic iodine (e.g. $CH_3I$, $CH_2ICl$, $CH_2I_2$) compounds
including those originating from marine algae (Saiz-Lopez and Plane, 2004) were considered
to be the primary source of iodine in the marine atmosphere (Carpenter, 2003; Vogt et al.,
1999). However, the biogenic sources of atmospheric iodine could not account for the levels
of IO detected in the tropical MBL (Mahajan et al., 2010b; Read et al., 2008). Currently,
inorganic iodine emissions are considered to be the dominant sources contributing to the open
ocean boundary layer iodine (Carpenter et al., 2013). A recent study by Koenig et al. (2020)
concluded that inorganic iodine sources play major role in comparison to the organic iodine
sources in contributing even to the upper troposphere iodine budget. Laboratory investigations
revealed that at the ocean surface, iodide ($I^-$) dissolved in the seawater reacts with the deposited



gas-phase ozone to release hypoiodous acid (HOI) and molecular iodine ($I_2$) via the following
reactions (Carpenter et al., 2013; Gálvez et al., 2016; MacDonald et al., 2014) :
$I^- + O_3 \rightarrow IOOO^-$ (R1)
$IOOO^- \rightarrow IO^- + O_2$ - " -
$IO^- + H^+ \leftrightharpoons HOI$ - " -
$H^+ + HOI + I^- \rightleftharpoons I_2 + H_2O$ (R2)
The reaction of sea surface iodide (SSI) with ozone in (R1) is considered a major contributor
(600-1000 Tg per year, (Ganzeveld et al., 2009)) to the loss of ozone at the surface ocean,
contributing between 20 % (Garland et al., 1980) and 100 % (Chang et al., 2004) of the oceanic
ozone dry deposition velocity. Reactions (R1) and (R2) result in the release of reactive iodine
(HOI and $I_2$) to the atmosphere, where they quickly photolyse to yield I atoms, which react
with ozone in the gas phase to form IO (Carpenter, 2003; Saiz-Lopez et al., 2012). Carpenter
et al. (2013) showed that the reactions (R1) and (R2) could account for about 75 % of the IO
levels detected over the tropical Atlantic Ocean. Further studies have shown that including
these reactions and the resulting fluxes of HOI and $I_2$ in atmospheric chemistry models has
results in good agreement between observed and modelled iodine levels over the Atlantic and
the Pacific Ocean, but not for the Indian and Southern Ocean. For example, the sea-air flux of
HOI and $I_2$ could explain the observed levels of molecular iodine and IO at Cape Verde (Lawler
et al., 2014b), and observed IO levels over the eastern Pacific were in reasonable agreement
with those modelled from estimated $I_2$ and HOI fluxes (MacDonald et al., 2014). In contrast,
the inorganic iodine fluxes estimated for the Indian Ocean and Indian sector of the Southern
Ocean marine boundary layer could not fully explain the observed IO concentrations (Mahajan
et al., 2019a, 2019b).



Predicted global emissions of iodine compounds show a large sensitivity (~ 50 %) to the SSI
field used (Saiz-Lopez et al., 2014; Sherwen et al., 2016a, 2016c); an improved and accurate
system for simulating SSI concentration is imperative. Existing global parameterisations
discussed in this study follow three different methods for SSI estimation. The first is a linear
regression approach against biogeochemical and oceanographic variables  (Chance et al.,
2014), the second uses an exponential relationship with sea surface temperature as a proxy for
SSI (MacDonald et al., 2014), and the third is a recent machine-learning-based model (Sherwen
et al., 2019a) that predicts monthly global SSI fields for the present-day. Where such
approaches are based on large scale relationships, they may not properly capture smaller scale,
regional differences in SSI (as observed for Chance et al., 2014; MacDonald et al., 2014) or
underestimate surface iodide concentration (in case of Sherwen et al., 2019).Furthermore, there
are large differences in predicted iodide concentrations between these parametrisations in some
regions (refer Sect. 3.2). Thus, estimation of seawater iodide based on the existing
parameterisations may not always be sufficiently accurate.
At present, there is a paucity of measurements of SSI, and remote sensing techniques cannot
detect iodine species in seawater (Chance et al., 2014; Sherwen et al., 2019a). In particular,
regions of the Indian Ocean and the Southern Ocean have been under-sampled in terms of
iodine observations in the atmosphere and ocean (Chance et al., 2014; Mahajan et al., 2019a,
2019b). It is important to remember that the most widely used parameterisation (MacDonald
et al., 2014) is built on a limited observational dataset from the Atlantic and Pacific Ocean
completely excluding the Indian Ocean and the Southern Ocean. The parameterisations
presented in Chance et al. (2014), are based on a larger data set including Southern Ocean
observations, but still only make use of two data points in the Indian Ocean. Furthermore, the
Sherwen et al. (2019) parameterisation uses the updated data set including the new Indian
Ocean SSI observations used in this study. Compounding the lack of Indian Ocean SSI



observations is the fact that parts  and in particular the Arabian Sea and the Bay of Bengal, do
not follow the same seasonal trends in salinity (D'Addezio et al., 2015) and sea surface
temperature (Dinesh Kumar et al., 2016) as each other on the same latitudinal band and hence
the currently used global iodide parameterisations in models i.e. MacDonald et al. (2014) may
not be appropriate for these areas. Here we use new SSI observations made as part of this study
(described in full in Chance et al. (2019b) and included in Chance et al. (2019a)) to test whether
the existing parameterisations can be directly applied to the Indian Ocean and if regional
specific parameterisations are more accurate compared to the former.
Although several measurements of IO have been reported around the globe (Alicke et al., 1999;
Allan et al., 2000; Frieß et al., 2001; Großmann et al., 2013; Mahajan et al., 2009, 2010a,
2010b; Prados-Roman et al., 2015), the remote open ocean still remains under-sampled. The
two documented observations of IO in the Indian Ocean and the Indian sector (Jan-Feb 2015
and December 2015) of the Southern Ocean were interpreted using parameterisations to
estimate the SSI concentrations in combination with observed ozone concentrations, to
subsequently calculate the resulting inorganic iodine fluxes. This approach suggested that the
observed atmospheric IO may not be well correlated with the inorganic fluxes and that biogenic
fluxes could play an important role (Mahajan et al., 2019a, 2019b). Here, we present
measurements of IO in the MBL of the Indian Ocean and the Southern Ocean during the 9[th]
Indian Southern Ocean Expedition (ISOE-9) conducted in January-February 2017, alongside
the first simultaneous SSI observations along the cruise track (Chance et al., 2019a). The iodide
observations were used to compute the inorganic iodine fluxes to compare with IO observations
along the cruise tracks. Further, observed SSI concentrations are used to compute region-
specific parameterisations for SSI concentrations, following the approaches taken by Chance
et al. (2014) and MacDonald et al. (2014). The iodide concentrations obtained with these
region-specific modified parameterisations are compared to the iodide estimates using their



original counterparts and the global machine-learning-based prediction of SSI concentration
(Sherwen et al., 2019a). The resulting estimated reactive iodine fluxes (HOI and $I_2$) are then
used to see if the inorganic fluxes can explain the IO loading in the atmospheric MBL.
**2. Measurement techniques and methodology**
The 9th Indian Southern Ocean Expedition (ISOE-9) was conducted from January to February
2017 in the Southern Ocean and the Indian Ocean sector of the Southern Ocean. The expedition
started from Port Louis, Mauritius, and spanned the remote open ocean area till the coast of
Antarctica. Observations of IO, SSI and $O_3$ were made along the cruise track during ISOE-9.
For further analysis we also include IO observations from the 2nd International Indian Ocean
Expedition (IIOE-2) and the 8th Indian Southern Ocean Expedition (ISOE-8) conducted in the
Indian and Southern Ocean region during austral summer of 2014-2015 (Mahajan et al., 2019a,
2019b). We also include SSI observations in the northern Indian Ocean from two expeditions
namely, the Sagar Kanya-333 cruise (SK-333) and the Bay of Bengal Boundary Layer
Experiment (BoBBLE) conducted during June-July and September 2016 respectively (Chance
et al., 2019b). Table 1 includes details of the expeditions, including the locations, dates of the
expeditions and the meridional transect for each expedition. Figure 1a shows a map with the
cruise tracks for the five expeditions. Figure 1b shows the seawater iodide sampling locations
during ISOE-9, SK-333 and BoBBLE expeditions. The track of the ship during ISOE-9 along
with the air mass back trajectories arriving at noon each day is given in the supplementary text
Fig. S1. The HYbrid Single-Particle Lagrangian Integrated Trajectory (HYSPLIT) model
(Rolph et al., 2017; Stein et al., 2015) was used to calculate the back trajectories. Similar back
trajectory plots and full cruise tracks for ISOE-8 and IIOE-2 are given in Mahajan et al. (2019a,
2019b). During the three expeditions, meteorological parameters of ocean and atmosphere were
measured using an on-board automatic weather station and manual observation techniques.





**2.1. Sea surface iodide (SSI)**
In this section, we focus on developing region-specific parameterisation for SSI estimation by
adapting previously established methods. The SSI concentrations obtained from the original
and newly developed region-specific parameterisation and SSI model predictions are used for
a comparison study, and further to calculate the inorganic iodine emissions.
**2.1.1 Observed SSI in the Indian Ocean and the Southern Ocean**
Historically, few observations of SSI are available for the Indian Ocean basin with reports of
only 3 data points in the open ocean from the Arabian Sea sector of the Indian Ocean
(Farrenkopf and Luther, 2002). Two of these values are coastal, and they lack supporting sea
surface temperature and salinity data; thus, they have been excluded from this study. However,
recent work has led to a large increase in the number of SSI observations available for the
Indian Ocean and Southern Ocean (Indian ocean sector) (Chance et al., 2019b). Specifically,
111 new observations were made during the 2016 ISOE-9 and 18 during the SK-333 and
BoBBLE. During the ISOE-9, SSI measurements in seawater were made concomitant with
observations of $O_3$ and IO in the gas phase for the first time. Observations of SSI made during
this expedition used the cathodic stripping voltammetry method with a hanging mercury drop
electrode as a working electrode (Campos, 1997; Luther et al., 1988). The seawater samples
were collected during the ISOE-9 at a 3-6-hour interval between 23° S and 70° S. Seawater
samples from the SK-333 cruise and BoBBLE were analysed following the same technique for
surface iodide concentrations. Iodide data from SK-333 and BoBBLE contributed to 18
additional data points between 10° N and 4° S making a total of 129 new locations (excluding
coastal and extremely high values above 400 nM; see Chance et al. (2019b) for details) for
observed SSI in the Indian Ocean and Southern Ocean region. This is a major sample size
compared to the global 2014 database (n=925) across all the global oceans (Chance et al.,



2014), and these data points contribute substantially to the recently updated iodide dataset
(Chance et al., 2019a) (n=1342). From here onwards, the iodide concentrations obtained from
sampling observations will be referred to as measured SSI as opposed to modelled SSI to
differentiate between the observed iodide concentrations and those calculated using the
parametrisations. All available observations made in the Indian Ocean basin as presented in
Chance et al. (2019a) have been included for the development of the region-specific
parameterisation presented in this work. Further details about the measurement technique and
the observations used can be found in Chance et al. (2019b).
**2.1.2 Iodide parametrisations**
Due to the sparsity of SSI measurements, different empirical parametrisations have been
proposed to estimate SSI concentrations. Parameters like SST and salinity (only for SK-333
and BoBBLE; $R^2$ = 0.3, P = 0.018) show a positive correlation with the SSI concentrations.
However, a global parameterisation scheme may not capture the specificities of these required
for regional studies. The northern Indian Ocean has markedly different sea surface salinity
(D'Addezio et al., 2015) and SST (Dinesh Kumar et al., 2016) in its two basins, the Arabian
Sea and the Bay of Bengal, that share the same latitude bands separated by the Indian sub-
continental landmass. These basins experience the biannually reversing monsoonal winds,
which greatly influence their SST and salinity structure. Strong winds in the Arabian Sea
associated with the summer monsoon dissipate heat via overturning and turbulent mixing.
Whereas weaker winds in the Bay of Bengal imply high SST due to the formation of stable and
shallow surface mixed layer (Shenoi, 2002). The Arabian Sea exhibits much higher salinity
compared to the Bay of Bengal due to greater evaporation and lower river runoff (Rao and
Sivakumar, 2003). Current global SSI parameterisations (MacDonald et al., 2014; Chance et
al., 2014) are based almost entirely on observations from the Atlantic, Pacific and Southern
(excluding the Indian ocean sector) Oceans, they may not be suitable for accurate estimation



of SSI in the distinct and highly variable salinity and temperature regimes of the Indian Ocean
region.
Here, we aim to create region-specific parameterisations for the Indian and Southern Ocean
and conduct a comparison between these and the existing global parameterisations, further
discussed in Sect. 4.2. The existing global and the new region-specific parameterisations are
listed in Table 2. Below we describe briefly the modified parameterisations. Details about the
original parameterisations can be found in their respective publications (Chance et al., 2014;
MacDonald et al., 2014; Sherwen et al., 2019a).
(a) Linear regression analysis was performed, on each parameter, namely, SST, mixed layer
depth (MLD), latitude, sea surface nitrate concentration, and salinity against the measured SSI
concentrations from ISOE-9, SK-333, and BoBBLE campaigns, similar to the Chance et al.
(2014) technique, but using in situ SST and salinity observations instead of climatological
values.  More details on the approach taken can be found in the supplementary text. The
combination with the largest $R^2$ and uniform distribution of residuals from the statistically
significant dependent variables, as detailed in Table S1 resulted in Eq. 2 of Table 2. Eq. 2 thus
represents a region-specific (the Indian Ocean and the Southern Ocean region abbreviated as
Ind. O. + Sou. O. in the figures) variant of the Chance et al. (2014) parameterisation for the
estimation of SSI concentrations. Similarly, keeping in mind the difference in the SST and
salinity for the Indian Ocean and the Southern Ocean, another parameterisation was derived
only for the Southern Ocean region using the ISOE-9 iodide observations and for the Indian
Ocean using the SK-333 and BoBBLE iodide observations, respectively. The parameterisation
for the Southern Ocean region using ISOE-9 iodide observations is given in Table 2 as Eq. (3).
A similar Indian Ocean parameterisation is not included in this text as the linear regression
analysis fails to obtain a parametric equation for this region. This may be due to fewer data
points (n=18) combined for the Arabian Sea and Bay of Bengal basins.





(b) A second method for the estimation of SSI concentration was proposed by MacDonald et
al. (2014) that uses the correlation between sea surface iodide and SST. At present, this is the
most commonly used parameterisation in global models (Sherwen et al., 2016c, 2016b, 2016a;
Stone et al., 2018). Similar to MacDonald et al. (MacDonald et al., 2014) (Table 2, Eq. 4), we
derived an Arrhenius-type, region-specific expression using iodide and SST data from ISOE-
9, SK-333 and BoBBLE. Figure 2 shows the typical linear dependence of $\ln[I^-]$, for observed
SSI in the Indian Ocean and the Southern Ocean, with $SST^{-1}$, which resulted in the Arrhenius
form expression given as Eq. (5) in Table 2.
Figure 3 shows the iodide concentrations for the three campaigns, ISOE-8, IIOE-2 and ISOE-
9, calculated using equations (1) to (5), the measured iodide concentrations from ISOE-9, SK-
333 and BoBBLE, and the global iodide model predictions obtained from Sherwen et al. (2019)
(Table 2.). From here on, region-specific parameterisations developed for SSI concentrations
are referred to as the modified versions of the original parameterisations; Eq. (2) and (3) are
called the modified Chance et al. (2014) parameterisation for the Indian Ocean and Southern
Ocean region and only the Southern Ocean region, respectively. Eq. (5) is called the modified
Macdonald et al. (2014) parameterisation. The machine-learning-based model proposed in
Sherwen et al. (2019) is referred to as 'SSI model' results.
**2.2. Ozone**
Surface ozone was monitored using a US-EPA approved nondispersive photometric UV
analyser (Ecotech EC9810B) installed on the ship during the expeditions to detect surface
ozone values at a one-minute temporal resolution. A Teflon tube ~ 4 m long fixed towards the
front of the ship acted as an inlet for the analyser. The analyser is equipped with a selective
ozone scrubber, which was alternatively switched in and out of the measuring stream. The
analyser has a lower detection limit of 0.5 ppbv and a precision of 0.001 ppmv. A 5-micron



PTFE filter membrane installed in the sample inlet tube was changed regularly. Zero and span
calibration were done every alternate day to ensure accurate $O_3$ measurements. The ozone data
collected was cleaned to remove the data points under the influence of the ship's smokestack
by referring to the measured apparent wind direction on the ship. Apparent wind approaching
the ship from 0 to 90º or 270 to 360º was considered free from smokestack emission influence,
where 0 or 360º represents the bow of the ship. Ozone data recorded when the ship was
stationary showed major smokestack emission influence and was excluded from the data.
**2.3. Estimation of inorganic iodine fluxes**
In order to estimate the contribution of inorganic iodine chemistry to active iodine chemistry
in the atmosphere, the atmospheric fluxes for the main product species, $I_2$ and HOI, need to be
calculated, since direct flux measurements of $I_2$ and HOI  have not been done anywhere in the
world to date. While there are reported observations of marine $I_2$ emission, they are few in
number and mostly from coastal regions (Atkinson et al., 2012; Huang et al., 2010; Saiz-Lopez
et al., 2006a) and one observation in the open ocean (Lawler et al., 2014a), although these are
all observations of atmospheric concentrations and not of fluxes. As observed SSI is not
available for all cruises, we used the following scenarios for SSI to estimate the inorganic
iodine fluxes:
(a)    Using measured SSI: Observations of sea surface iodide from ISOE-9, SK-333, and

BoBBLE.

(b)    Using calculated SSI from:

1. Chance et al. (2014) parameterisation                                    Eq. (1)

2. Modified Chance et al. (2014) parameterisation for the Indian Ocean and

Southern Ocean (Ind. O. + Sou. O.) region                              Eq. (2)





3. Modified Chance et al. (2014) parameterisation for the Southern Ocean (Sou.

O.) region                                                            Eq. (3)

4. MacDonald et al. (2014) parameterisation using SST                     Eq. (4)
5. Modified MacDonald et al. (2014) parameterisation                      Eq. (5)
6. Using machine learning SSI model predictions (Sherwen et al., 2019a) Eq. (6)
Ozone was measured on all three cruises (ISOE-9, IIOE-2 and ISOE-8). The fluxes for HOI
and $I_2$ were then calculated for all the above scenarios except for the observations from SK-
333 and BoBBLE as IO observations were not taken during these cruises. The following
algorithm was used for estimating iodine fluxes (Carpenter et al., 2013),
$flux_{I_2} = \left[O_{3\,(g)}\right] * \left[I^-_{(aq)}\right]^{1.3} * (1.74 \times 10^9 - 6.54 \times 10^8 * \ln(ws))$          Eq. (7)
$flux_{HOI} = \left[O_{3\,(g)}\right] * \left(4.15 \times 10^5 * \dfrac{\sqrt{[I^-_{(aq)}]}}{ws} - \dfrac{20.6}{ws} - 2.36 \times 10^4 * \sqrt{[I^-_{(aq)}]}\right)$ Eq. (8)
where, the fluxes are in nmol m$^{-2}$ d$^{-1}$, [$O_3$] in nmol mol$^{-1}$ (ppbv), [I$^-$] in mol dm$^{-3}$ and the wind
speed (WS) in m s$^{-1}$. Carpenter et al. (2013) did not consider the effect of temperature in the
interfacial layer of the sea-surface model on activation energies for the reaction R1 (i.e.,
assumed the temperature dependence for k (I$^-$ + $O_3$) to be zero). Although $I_2$ and HOI fluxes
are expected to increase with the temperature of the interfacial layer, $I_2$ production has a
negative activation energy, as noted by MacDonald et al. (2014).  In Carpenter et al. (2013)
(specific to the tropical Atlantic), a seawater temperature of 15ºC and air temperature of 20º C
were used to calculate Henry's law constants, diffusion constants, and mass transfer velocities.
Again assuming the temperature dependence of k(I$^-$ + $O_3$) to be zero, but including the
temperature-dependence of Henry's law constants, diffusion constants, and mass transfer
velocities, the same interfacial layer model predicted effective activation energies for $I_2$ and
HOI emissions of $-2$ kJ mol$^{-1}$ and 25 kJ mol$^{-1}$ (Macdonald et al. (2014).  Using these



activation energies, Macdonald et al. (2014) calculated differences in $I_2$ and HOI fluxes of 3 %
and 31-41 %, respectively, at SSTs of 10º C and 30º C compared to the room-temperature
parameterisations presented in Carpenter et al. (2013). Experimentally derived activation
energies for $I_2$ and HOI emissions were $-7 \pm 18$ kJ mol$-1$ and $17 \pm 50$ kJ mol$^{-1}$ (MacDonald
et al., 2014). As HOI represents the larger iodine flux, the higher relative uncertainty in the
activation energy should be kept in mind when calculating temperature-dependent emissions.
HOI and $I_2$ fluxes are also influenced by the wind speed as seen from equations (7) and (8),
and the modelled iodine fluxes (HOI and $I_2$) are highest for high $[O_3]$, high $[I^-]$ and low wind
speed. This is explained by the assumption that wind shear drives mixing of the interfacial layer
to bulk seawater, reducing the efflux of HOI and $I_2$ into the atmosphere (Carpenter et al., 2013).
Negative fluxes are obtained from equations (7) and (8) for both HOI and $I_2$ when the wind
speed is higher than 14 m s$^{-1}$, which is not physically possible and therefore the model output
is limited to wind speeds below 14 ms$^{-1}$ (Mahajan et al., 2019a). Iodine fluxes calculated from
equations (7) and (8) using SSI concentrations from the scenarios (a) and (b 1-6) are shown in
Fig. 4 (c and d).
**2.4. Iodine Oxide**
**2.4.1 Observations**
Ship-based measurements of IO were made using the Multi-Axis Differential Optical
Absorption Spectroscopy (MAX-DOAS) technique (Hönninger et al., 2004; Platt and Stutz,
2008). The MAX-DOAS was installed at the bow of the ship with a direct line of sight towards
the front of the ship to avoid the ship's plume in the detection path of the telescope. The MAX-
DOAS was programmed to capture scattered sunlight spectra at every 1 second at set elevation
angles of 0, 1, 2, 3, 5, 7, 20, 40, and 90-degrees during daylight hours. Mercury line calibration
offset, and dark current spectra were recorded after sunset on each day. Elevation angles





outside a range of ±0.2 degree from the set value were eliminated from the 30 minutes averaged
spectra for increased accuracy. Figure S2 shows the resultant IO and $O_4$ differential slant
column densities (DSCDs) for ISOE-9 campaign, similar plots are available for ISOE-8
(Mahajan et al., 2019a) and IIOE-2 (Mahajan et al., 2019b). The QDOAS software (Danckaert
et al., 2017) was used for DOAS retrieval of IO from the spectra using the optical density fitting
analysis method. The spectra were fitted with a $3^{rd}$ order polynomial using fitting interval of
415 to 440 nm with cross-sections of  $NO_2$ (Vandaele et al., 1998), $O_3$ (Bogumil et al., 2003),
$O_4$ (Thalman and Volkamer, 2013), $H_2O$ (Rothman et al., 2013), two ring spectra, first as
recommended by Chance and Spurr, (1997) and second following Wagner et al.,( 2009) and a
liquid water spectrum for seawater (Pope and Fry, 1997). To remove the influence of
stratospheric absorption a spectrum corresponding to 90º (zenith) from each scan was used as
a reference for the analysis. The raw spectra were analysed to obtain differential slant column
densities (DSCDs), and values with a root mean square error (RMS) of greater than $10^{-3}$ were
eliminated. Similarly, DOAS retrieval of $O_4$ in 350 to 386 nm spectral window was performed,
and DSCDs were obtained. The optical density fits for IO and $O_4$ from ISOE-9 are shown in
Fig. S3. The IO DSCDs were then converted to volume mixing ratios using the $O_4$ slant
columns following the previously used "$O_4$ method" (Mahajan et al., 2012; Prados-Roman et
al., 2015; Sinreich et al., 2010; Wagner et al., 2004). Further details of the instrument, retrieval
procedure and conversion into mixing ratios can be found in previous works (Mahajan et al.,
2019a, 2019b).
**2.4.2 Modelled atmospheric IO**
We use outputs from two global models for a comparison with the observations conducted
during the three cruises. The first model is the GEOS-Chem chemical transport model (version
10-01, 4x5 degrees horizontal resolution, http://www.geos-chem.org,), which includes detailed
HOx-NOx-VOC-ozone-halogen-aerosol tropospheric chemistry (Sherwen et al., 2016c, 2017)



and is driven by offline meteorology from NASA's Global Modelling and Assimilation Office
(http://gmao.gsfc.nasa.gov) forward processing product (GEOS-FP).
The second model is the 3D chemistry-climate model CAM-Chem version 4 (Community
Atmospheric Model with Chemistry)  https://www2.acom.ucar.edu/gcm/cam-chem), which is
included in the CESM framework (Community Earth System Model, CAM-Chem, version
4.0). The model includes a state-of-the-art halogen chemistry scheme (chlorine, bromine and
iodine) (Saiz-Lopez and Fernandez, 2016). The current configuration includes an explicit
scheme of organic and inorganic iodine emissions and photochemistry. These halogen sources
comprise the photochemical breakdown of five very short-lived bromocarbons ($CHBr_3$,
$CH_2Br_2$, $CH_2BrCl$, $CHBrCl_2$ and $CHBr_2Cl$) naturally emitted by phytoplankton from the
oceans (Ordóñez et al., 2012). The model was run in specified dynamic mode (Ordóñez et al.,
2012), with a spatial resolution of 1.9° latitude by 2.5° longitude and 26 vertical levels from the
surface to up to 40 km.
Both models include biotic emissions of four iodocarbons ($CH_3I$, $CH_2ICl$, $CH_2IBr$ and $CH_2I_2$)
as described by (Ordóñez et al., 2012) and abiotic oceanic sources of HOI and $I_2$ based on the
Carpenter et al. (2013) and MacDonald et al. (2014) laboratory studies of the oxidation of
aqueous iodide by atmospheric ozone at the ocean surface. Both models here use the
MacDonald parameterisation expression (Eq. (4), MacDonald et al., 2014) discussed in Section
2.1.2 to predict surface iodide used for calculating iodine emissions and the organo-halogen
emissions from Ordoñez et al. (2012). IO surface concentrations for the three campaigns (IIOE-
2, ISOE-8 and ISOE-9) were extracted from the model runs and used for comparison.
Currently, these two global models include reactive iodine chemistry (along with TOMCAT,
which includes the tropospheric iodine chemistry (Hossaini et al., 2016)).
**3. Results**



### 3.1 Ozone, Meteorological and Oceanic parameters

The latitudinal distribution of hourly average values of wind speed (WS), $O_3$, SST, and salinity from all the campaigns are shown in Fig. 5. Winds arriving at the ship, shown in the first panel (Fig. 5a), remained low for most of the duration of all three expeditions with wind speed ranging from 1 m s$^{-1}$ to stronger winds of 24 m s$^{-1}$ on a few days. Even stronger winds (above 30 m s$^{-1}$) were observed during the ISOE-9 in the region between 64 ° and 65º S with the highest wind speed of 32 m s$^{-1}$ at 66º S on the night of 8$^{th}$ February 2017. Ozone mixing ratios, (Fig. 5b) during all three expeditions showed a similar trend exhibiting a large reduction in values in the open ocean environment compared to coastal environments. The back trajectories (supplementary text) show that for most of the expeditions, air masses arriving at the cruise were from the open ocean environment and did not have any anthropogenic influence for the last five days. This is reflected in the $O_3$ values, which range between 8 and 20 ppbv in the open ocean but were between 30 and 50 ppbv near the coastal regions, where the air mass back trajectories confirm anthropogenic origins. Close to the Indian sub-continent ozone levels peaked at about 50 ppbv during the ISOE-8. It also showed a distinct diurnal variation with higher ozone values during the daytime due to photochemical production. However, in the open ocean environment, ozone mixing ratios did not show this diurnal variation, and indeed values of ozone dropped during daytime indicating photochemical destruction during both ISOE-8 and ISOE-9 (Fig. 5b).

As already noted, SST is widely used to predict SSI (Eq. 4 and 5). Combined SST data (Fig. 5c) reveal a steady decrease in sea surface temperature from 15º S to 68º S for all the campaigns. During January 2015 (ISOE-8) seawater north of 6º N displays slightly lower SST (~ 3º C) compared to that in December 2015 (IIOE-2). Salinity is also an important parameter for the prediction of SSI (higher coefficient in Eq. 1, 2 and 3). The Southern Ocean region explored during ISOE-8 and ISOE-9 reveals similar salinity values (Fig. 5d) for the austral



summer months of 2014 and 2016 (January-February). The salinity data shows relatively lower
values for ISOE-8 compared to those for IIOE-2 for the region 15º N to 20º S. Despite the inter-
annual differences in the northern Indian Ocean region, salinity values of ~ 35 PSU overlap for
the IIOE-2 and ISOE-8 in a small window of 7º N to the equator. Below the equator, the salinity
values for IIOE-2 increase while for ISOE-8 salinity remains lower than 35 PSU until 20º S.
Seawater between 20º S and 44º S has a near-constant salinity of 35 PSU which decreases to
~33.5 PSU after 44º S and remains the same until 65º S after which the salinity begins to drop
to 31.5 PSU near 67º S close to Antarctica.

**3.2 Sea surface iodide concentration**

Latitudinal averages of SSI concentrations estimated from seven scenarios (listed in Sect. 2.3)
are shown in Fig. 3. SSI estimates from the IIOE-2 campaign are marked separately to
differentiate from the ISOE estimates for the Indian Ocean region. There is a clear difference
in the estimated SSI in different scenarios. All the estimates and the model follow a similar
pattern showing elevated levels in the tropics as compared to the higher latitudes. SSI estimates
from parameterisations (Eq. 1, 3, 4, and 5) show nearly constant values for SSI from 15º N to
25º S, after which a steady decline is noted until 70º S. Thus, the parametrisations based on Eq.
1, 3, 4 and 5 do not capture the decreasing trend observed for iodide around the equator. Eq. 2,
which was derived specifically for the Indian Ocean and Southern Ocean region better captures
this trend, and also shows a better match with the measured SSI from SK-333 and BoBBLE in
the Indian Ocean. Eqn. 6 also predicts lower concentrations around the equator than in the
northern Indian Ocean. SSI concentrations estimated using the Chance et al. (2014)
parameterisation (Eq. 1) show a small increase in iodide concentrations south of 47º S (polar
front), which is not observed in the other parameterisations, but there is some suggestion of in
the observations. Eq. 1 also resulted in a large difference (~ 50 nM) of SSI estimates north of
10º N between the IIOE-2 and ISOE-8 cruises; while this difference was lower for the other



parameterisations. This difference between the SSI estimates for the IIOE-2 and ISOE-8 cruises
is due to the large difference in salinity values for this region (Sect. 4.1). SSI estimates using
Eq. 2 shows good agreement with the model prediction of Sherwen et al. (2019), both
indicating a decrease in SSI concentrations near the equator during the IIOE-2 and ISOE-8
expeditions. Some high SSI concentrations (up to ~250 nM) were observed around 10° N, these
were best replicated by Eqn.3. The highest SSI concentrations estimated using Eq. 3 were 244
nM at 7° N during IIOE-2 and 242 nM at 12° S during ISOE-8. At the equator, Eq. 2 performs
better in predicting the SSI concentrations with a difference of ~75 nM compared to the
observations. SSI estimates from Eq. 4, i.e. MacDonald et al. (2014) parameterisation, were
lower than the measured iodide concentrations and all other parameterisation, including the
model (Eq. 7) predictions. Overall, all modified parameterisations (Eq. 2, 3 and 5) estimate
higher SSI compared to the original parameterisation (Eq. 1 and 4), with the exception of the
region south of 20° S, where Eq. 3 predicts lower SSI than Eq. 1. The modified MacDonald
parameterisation (Eq. 5) estimated iodide concentrations to be greater by 50 nM for the entire
dataset in comparison to the existing MacDonald parameterisation given by Eq. 4. For Eq. 5,
the uncertainty in the iodide concentration from the 95 % prediction band is ~15 % of the
predicted value.
**3.3 Iodine fluxes**
Figure 4 shows the latitudinal variation in IO mixing ratios, inorganic iodine emissions (HOI
and $I_2$), chl-*a* and ozone mixing ratios for the entire dataset comprising of the three campaigns.
All the panels in Fig. 4 are plots of daily averaged values during each expedition, except for
the HOI and $I_2$ fluxes; these are latitudinal averages from each campaign. Emissions calculated
using the measured SSI concentrations (represented by filled spheres in Fig. 4 c & d) from
ISOE-9 correspond to the data points of the measured SSI concentration. Oceanic inorganic
iodine emission fluxes of HOI and $I_2$ were estimated using the Carpenter et al. (2013)





parameterisation given in Eq. (7) and (8) limited to wind speeds below 14 m s$^{-1}$. Thus, the
fluxes estimated from the measured SSI concentrations were reduced to 56 points (out of 111
measured SSI data points). The seven different datasets of iodide concentrations (listed in Sect.
2.3) have been used for estimation of HOI and I$_2$ fluxes. For the entire dataset, the highest
fluxes were obtained when using the SSI concentrations from the modified Chance et al. (2014)
parameterisation (Eq. 3), derived from measured SSI from the Southern Ocean region, i.e.
during ISOE-9. The second highest fluxes were estimated using SSI from Eq. 2, obtained from
measured SSI from the Indian Ocean and Southern Ocean. Comparatively lower iodine
emissions were estimated using SSI concentration from MacDonald et al. (2014)
parameterisation (Eq. 4). The estimated inorganic iodine fluxes in the Southern Ocean region
(30º S and below) are much lower compared to the Indian Ocean (Fig. 5), driven by the higher
estimated SSI in the latter. Maximum inorganic emissions are predicted in the tropical region,
specifically, north of the equator. HOI is the dominant reactive iodine precursor species for the
entire dataset, with calculated flux values 20 times higher than those for I$_2$. Emissions estimated
using SSI from Eq. (3), resulted in a peak HOI flux of $1.5 \times 10^9$ molecules cm$^{-2}$ s$^{-1}$ at 9º N during
ISOE-8. The lowest HOI flux of $1.7 \times 10^6$ molecules cm$^{-2}$ s$^{-1}$ was obtained at 61º S during ISOE-
9. For the same latitudes (9º N and 61º S), a maximum I$_2$ flux of $7.0 \times 10^7$ molecules cm$^{-2}$ s$^{-1}$
and a minimum of $1.3 \times 10^5$ molecules cm$^{-2}$ s$^{-1}$ were estimated, respectively. Flux estimates from
Eq. 2 are slightly lower, with a maximum HOI flux of $1.3 \times 10^9$ and a minimum of $5.8 \times 10^5$
molecules cm$^{-2}$ s$^{-1}$ and maximum I$_2$ flux of $5.2 \times 10^7$ with minimum of $8.3 \times 10^4$ molecules cm$^{-2}$
s$^{-1}$ at the same latitudes. The estimated HOI and I$_2$ emissions are notably lower (by ~50 %)
during IIOE-2 to the north of 5ºS compared to emissions from ISOE-8. Between 5º S and 20º
S, the emissions from IIOE-2 and ISOE-8 are similar. Fluxes estimated using measured SSI
concentrations for the ISOE-9 campaign (20º S to 70º S) show no strong latitudinal trend for
both HOI and I$_2$ emissions. The maximum calculated HOI flux was $5.8 \times 10^8$ molecules cm$^{-2}$ s$^{-}$


$^1$ at 68° S and the minimum was $1.1\times10^7$ molecules cm$^{-2}$ s$^{-1}$ at 33° S. Similarly, I$_2$ fluxes
estimated from measured SSI concentrations peaked at $1.5\times10^7$ molecules cm$^{-2}$ s$^{-1}$ at 32° S with
a minimum of $3.5\times10^5$ molecules cm$^{-2}$ s$^{-1}$ at 67° S. Inorganic iodine emissions estimated using
model predictions for SSI concentrations from Sherwen et al. (2019) match well with the fluxes
estimated using the iodide parametrisation tools. Despite the differences in SSI concentrations
from existing and region-specific parameterisations, all result in similar values for iodine fluxes
and so SSI cannot explain discrepancies in the observed and modelled IO levels in this region.
**3.4 Iodine oxide**
**3.4.1 Observations**
IO was detected above the instrument detection limit ($2.1 - 3.5 \times 10^{13}$ molec. cm$^{-2}$ i.e. $0.4 – 0.7$
pptv) in all three campaigns. The expeditions covered a track from the Indian Ocean to the
Antarctic coast in the Southern Ocean and showed lower IO DSCDs in the tropics compared
to the Southern Ocean, with a peak of about $3 \times 10^{13}$ molec. cm$^{-2}$ at 40° S. Figure 4a shows
daily averaged IO mixing ratios for all the three cruises combined. IO mixing ratios of up to 1
pptv were observed in the region 50° - 55° S and slightly higher values of IO mixing ratios were
observed in the region below 65° S close to the Antarctic coast. North of the polar front region,
the maximum IO average mixing ratio of ~1 pptv was observed at 40° S. The highest values of
IO were observed close to the Antarctic coast, with up to 1.5 pptv measured during ISOE-9
and similar values are reported for the ISOE-8 expedition south of the polar front (Mahajan et
al., 2019a). The IO mixing ratios in the Southern Ocean region for ISOE-9 ranged between 0.1
and a maximum of 1.57 ($\pm$ 0.37) pptv observed on 18 Feb 2017 at 50° S on a clear sky day.
This maximum value was observed only on one day, and preceded by foggy and misty days,
later followed by overcast for several days evidencing the role of photochemistry in IO
production from its precursor gases.



### 3.4.2 Modelled IO

Based on the current understanding of iodine chemistry, regional and global models consider inorganic fluxes of iodine (HOI and $I_2$) as major contributors of iodine in the marine boundary layer. It is important to verify if the models using the existing parameterisation for these source gases can replicate observations of IO in the region of study. Thus, we have included model IO output from GEOS-Chem and CAM-Chem, both of which use the SST based MacDonald et al. (2014) parameterisation for SSI (Fig. 4b). The surface IO output from GEOS-Chem predicts the highest levels of IO up to 1.7 pptv to the north of the equator at 11º N for the time period of the IIOE-2 campaign. For the same latitudes, the model suggests lower IO levels, of less than 0.5 pptv, during the ISOE-8 campaign. Conversely, south of the equator to 10º S, the model predicts higher IO levels during the ISOE-8 and lower IO values during the IIOE-2, in agreement with the observations. Below 10º S, IO predictions for both campaigns match well until 20º S, which was the latitudinal limit for the IIOE-2 campaign. To the south of 20º S, modelled IO levels remained below 1 pptv and exhibited a decreasing trend to the south of the polar front, in disagreement with IO observations. At locations between 40º S and 43º S, GEOS-Chem underestimates the observed IO levels by 50 %. These locations are close to the Kerguelen Islands, and high IO values were observed here only during the ISOE-8. These locations have been omitted in the correlation study between modelled and observed IO as they could be impacted by coastal or upwelling emissions, which are not well prescribed in the models.

The CAM-Chem IO surface output suggests consistently higher levels of IO during IIOE-2 compared to the ISOE-8 for the same latitudinal band (Fig. 4b). Contrary to the observations, the CAM-Chem model suggests that IO levels during the IIOE-2 are up to 1 pptv higher than the ISOE-8 campaign near 7º S latitude. The model also shows elevated IO levels of 2.7 pptv at 7.9º N during the IIOE-2 campaign, which does not match the observations during the IIOE-

2 or the ISOE-8 for that region. IO levels below 1.5 pptv (11º N to 20º S) are indicated for the
ISOE-8 campaign. In addition, the region between 0º and 1.5º S has similar IO levels for the
IIOE-2 and ISOE-8 campaigns. The model predicts lower IO levels for the south Indian Ocean
and the Southern Ocean (less than 1 pptv) with decreasing IO to the south of the polar front.
However, at 43º S, the model suggests higher IO (2.4 pptv) during the ISOE-9, which matches
the increase in observed IO for that region during the ISOE-8 expedition, with this region being
close to the Kerguelen Islands Both models show consistently higher absolute concentrations
overall compared to the observations north of the polar front.
**4. Discussion**
**4.1 Seawater iodide**
To improve the estimation of SSI in the study region, previously established parameterisations
(Eq. 1 and 4) were modified to obtain a region-specific parameterisation for SSI concentrations.
SSI estimated using these modified parameterisations were less sensitive to seasonal salinity
and SST changes for the north Indian Ocean basin compared to the existing parameterisation
(Fig 3). Figure 6 shows the correlations of all the calculated SSI concentrations with the
observations. The SSI estimates from Eq. 1 to 6 correlate positively (significantly) to the
measured SSI concentrations (observations) from ISOE-9 (Fig. 6). Out of the six
parameterisation tools compared in this study, as expected, SSI from Eq. (2) i.e. the modified
Chance equations for the Indian Ocean and the Southern Ocean showed the best correlation
with the measured SSI because they were created using datasets from these campaigns (Fig. 6
and Table 2). Although the region-specific parameterisations were expected to match with the
observations they are based on, there was a notable difference between predictions and
observations when this approach was applied only to Indian Ocean SSI measurements from
SK-333 and BoBBLE ($R^2 = 0.5$ for Indian Ocean parameterisation, analysis not shown). This





could be attributed to the lack of SSI measurements in this region (n=18), and it highlights the
fact that there may be not only seasonally but regionally varying complexities in SSI which
should be considered when estimating SSI. All parameterisation methods used for SSI
estimations show that SSI concentrations are directly proportional to seawater salinity (listed
in Sect. 2.3). It is evident from Fig. 5d and Fig. 3a that to the north of the equator, the
parameterisations (Eq. 1 to 5) show lower SSI concentrations in regions with lower salinity (up
to 5° N during ISOE-8 – filled symbols Fig. 3) and higher SSI concentrations in regions with
comparatively higher salinity (during IIOE-2 – unfilled symbols Fig. 3). Only the modelled
SSI concentrations using Eq. 6 (Fig. 3a, data in purple) reveal an inversely proportional
relationship for salinity and SSI concentration in this region. The Sherwen et al. (2019)
parametrisation (Eq. 6) produces lower SSI concentrations in high salinity Arabian Sea waters
during IIOE-2 (Fig. 3a) north of 5° N, compared to the low salinity Bay of Bengal waters during
ISOE-8 which contradicts all the other parameterisation (Eq. 1 to 5). Further, the SSI
concentrations obtained from Sherwen et al. (2019) reverse their trend to the south of 6° N,
with higher concentrations during IIOE-2 and lower during ISOE-8. It should be noted that
only a few observations of SSI exist in this region to confirm this trend. Further discussion on
the relationship between salinity and other biogeochemical variables with SSI concentrations
at a global and regional scale can be found elsewhere (Chance et al., 2014, 2019a).
SSI estimates considering only SST as a proxy for iodide concentration (Eq. 4), reveal positive
correlations with measured SSI concentration (R = 0.86, P<0.001, n = 129; Fig. 6d). The
modified MacDonald parameterisation (Eq. 5) also correlates positively to the measured SSI
concentration but has a slightly lower coefficient of correlation (R = 0.83, P<0.001, n = 129;
Fig. 6e). When using the SST as a proxy for SSI, a large intercept was obtained for the SSI
values, evidencing the discrepancy in absolute value between this parametrisation and the
observations. Eq. (5) resulted in a lower intercept, approximately half of that for Eq. (4), and a





lower absolute slope value of |-3763±218| compared to the |-9134±613| of Eq. (4) given in
MacDonald et al. (2014). The lower absolute slope value for Eq. (5) implies that the SSI
concentrations for this region were less sensitive to the changes in SST compared to that in Eq.

(4).

Despite the lower R-value, the SSI estimates from Eq. 5 in Fig. 3 are closer to the measured
SSI concentration than the estimates from Eq. 2 and 3 for the region from 25º S to 70º S.
However, north of 25º S, the SSI estimates from Eq. 3 and Eq. 5 differ by ~40 %. Both SST
based parameterisation (Eq. 4 and 5) did not show the observed latitudinal variation in the SSI
concentrations near the equator. Linear regression of SSI with SST for only the Indian Ocean
region revealed that there was no correlation between the two ($R^2 = 0.07$, P = 0.3, n = 18). The
SSI in this region only showed dependence on the salinity and latitude, correlations with the
other parameters were not significant. This highlights that SST may not be a very good proxy
for SSI in the Indian Ocean, especially near the equator.  This is explored further in Chance et
al. (2019b). The original Chance et al. (2014) parameterisation displays higher sensitivity to
seasonal salinity changes compared to the existing and modified parameterisation in the Indian
Ocean region (Sect. 3.3). However, this method predicted increasing iodide concentration to
the south of the polar front (47º S), which is not supported by observations in this region (Fig.
3). In conclusion, considering the correlation with measured SSI concentration and dependence
on seawater salinity, the region-specific modified Chance parameterisation (Eq. 2) is a suitable
method to estimate SSI concentration for the Indian Ocean and Southern Ocean region. The
modelled SSI estimates by Sherwen et al. (2019) capture SSI trend close to equator better than
other existing schemes but it fails to replicate higher SSI observations at locations 8º N, 40º S
and to the south of 65º S close to the Antarctic coast (Fig. 3).
**4.2 Atmospheric iodine**





Combined IO observations from IIOE-2, ISOE-8, and ISOE-9 (Fig. 4a) show that the Indian
Ocean region has comparatively less IO in its MBL than the Southern Ocean region. IO
remained below 1 pptv up to 40º S and reached a maximum IO of 1.6 pptv south of the polar
front. Modelled surface IO output from GEOS-Chem and from CAM-Chem using the
Macdonald et al. (2014) parameterisation (Fig. 4b) do not match the observations of IO,
although they generally show good agreement with each other. The models show similar spatial
patterns across the entire dataset, except for two periods of very high IO levels predicted by
CAM-Chem (Fig. 4b). As well as structural differences between CAM-Chem and GEOS-
Chem, there are many halogen specific differences in rate constants, heterogeneous
parameters, cross-sections and photolysis of species (e.g. higher iodine oxides) which could
explain differences in predicted gas-phase IO. Considering the generally lower wind speeds
and higher ozone concentrations seen in IIOE-2 versus SOE-8 and SOE-9, the calculated fluxes
are higher and therefore more sensitive to assumptions, such as minimum wind speeds provided
to the Carpenter et al. (2013) parameterisation.  GEOS-Chem uses a minimum wind speed of
$5 \text{ m s}^{-1}$; however, CAM-Chem uses a minimum wind speed of $3 \text{ m s}^{-1}$.
Both models suggest higher than observed IO levels in the Indian Ocean region but under-
predict IO for the Southern Ocean region. The highest detected IO levels, both in the Southern
Ocean and in a narrow band around 43º S, were not reflected in the model predictions. We note
these occurred in regions of elevated chl-*a* values (Fig. 5), and that Mahajan et al. (2019a) also
reported positive correlations for IO with chl-*a* for the Indian Ocean region, above the polar
front for a subset of the dataset (ISOE-8). Calculated fluxes of HOI and $I_2$ (Fig. 4c and d) fail
to directly explain trends in the detected IO levels for the entire dataset, regardless of the
method used to estimate SSI. Maximum levels of HOI and $I_2$ predicted to the north of 5º N
correspond to rather low levels of IO (< 0.5 pptv) in this region. However, this has been
attributed to $NO_x$ titration of IO (Mahajan et al., 2019b). The models, however, do not capture



this iodine titration by $NO_x$ as seen in the observations; even though the reactions of IO with
NOx are included (Ordóñez et al., 2012). Similarly, for the region south of the polar front, the
calculated iodine fluxes remain low in the region of the maximum detected IO concentrations
during the ISOE-8 and ISOE-9 campaigns. Iodine fluxes estimated for the Indian Ocean region
(15º N to 5º N) during IIOE-2 and ISOE-8 show large differences with much higher values
during ISOE-8. However, the modelled IO is in fact higher for IIOE-2 than during ISOE-8 (5º-
15ºN). Considering that the models do not reflect the fluxes, this indicates that photochemistry
led to this difference in the model. Additionally, the elevated levels of IO predicted in the
models suggest that CAM-Chem and GEOS-Chem overestimate the impact of iodine chemistry
in the northern Indian Ocean.
In Fig. 7, correlations of iodine fluxes estimated using the measured SSI concentrations (Eq.
2) show that fluxes of HOI correlate positively with tropospheric ozone (R = 0.56, P<0.001)
and negatively to wind speed (R = -0.62, P<0.001) and $I_2$ fluxes correlate positively with SSI
concentration (R = 0.56, P = P<0.001) and ozone (R = 0.59, P<0.001) and negatively to wind
speed (R = -0.4, P<0.001).  This indicates that although there is positive correlation of $I_2$ with
SSI, the dominant inorganic iodine flux i.e. HOI does not show significant correlation with SSI
concentration, although the flux equation includes an iodide term (Eq. 8). We analysed the
correlation of daily averaged observed IO during the three campaigns with daily averaged
values of oceanic parameters (SST, chl-$a$, salinity, SSI concentration), meteorological
parameters (wind speed, ozone) and calculated inorganic iodine fluxes. We divided the
combined dataset from three campaigns into two regional subsets for the north (Fig. 8a) and
south (Fig. 8b) of the polar front (47º S). The correlation for SSI concentrations is included for
all the seven methods for SSI estimation listed in Sect. 2.3. The fluxes of HOI and $I_2$ obtained
using the seven different datasets for SSI are included and listed in Fig. 8 in the same order as
the SSI concentration (labelled 1 to 7). IO model output from GEOS-Chem (labelled 8) and





CAM-Chem (labelled 9) is included for the correlation analysis, along with chl-*a* data from
observations during ISOE-8 and ISOE-9 and satellite dataset obtained from MODIS Aqua
(Oceancolor, NASA-GSFC, 2017).
For the entire dataset (Fig. 8c), only wind speed shows a statistically significant, positive
correlation with observed IO above the 99 % confidence limit (R = 0.4, P<0.001, n = 115).  A
similar positive correlation with wind speed was found in the subset of data south of the polar
front (Fig. 8b) (R = 0.49, P = 0.01, n = 48), with observations north of the polar front showing
a weaker positive correlation (R = 0.27, P = 0.08, n = 67). Mahajan et al. (2012) showed that
no correlation existed between IO and wind speed over the eastern Pacific Ocean, contrary to
the results in this study. Current estimation methods for iodine emissions have a negative
dependence on wind speed (Eq. 7 and 8). A positive correlation of IO with wind speed could
suggest that increased vertical mixing enables emission of HOI and $I_2$, and/or other iodine
gases, thus enhancing IO production in the MBL. However, the interfacial model still over
predicts IO concentrations at low wind speeds due to over prediction of HOI and $I_2$ emission
(MacDonald et al., 2014). The apparently contradictory results from different studies call for
more observations of IO in the MBL over a range of wind speeds.
Salinity and SST show a weak negative correlation with atmospheric IO for the entire dataset
and for the north of the polar front region. This indicates that even if the physical parameters
are significant for the initial parametrisation for SSI and inorganic flux estimation, there is no
direct and significant correlation of these parameters with the atmospheric IO. However, south
of the polar front, SST correlates positively above the 99 % limit (R = 0.52, P = 0.01, n = 48)
and salinity correlates positively above the 95 % limit (R = 0.44, P = 0.03, n = 48).  Ozone
correlates negatively with IO above 95 % limit (R = -0.4, P = 0.046, n = 47), which could
indicate catalytic destruction of tropospheric ozone through atmospheric iodine cycling in the



south of the polar front. This highlights that although these physical parameters may be
required for iodine fluxes, IO levels may only be weakly related to them.
The calculated SSI concentrations and the HOI and $I_2$ fluxes calculated using these SSIs all
show a significant negative correlation with the observed IO concentrations above the 95 %
confidence limit for the entire dataset (except for the HOI flux estimated from the MacDonald
et al. (2014) parameterisation, which shows no significant correlation). The positive correlation
of the observed IO with wind speed is a potential driver for the negative correlation of observed
IO with the calculated HOI and $I_2$ fluxes, which decrease with wind speed.
Measured iodide levels (labelled 4) and the $I_2$ and HOI fluxes calculated from them (also
labelled 4) show no correlation with the observed IO levels across the entire dataset, although
iodide shows a significant positive correlation (R = 0.55, P = 0.04, n = 32) for IO measured
south of the polar front. Mahajan et al. (2019a) pointed out that SST negatively correlated with
IO for the ISOE-8 campaign, contradicting the previous results for observations in the Pacific
Ocean (Großmann et al., 2013; Mahajan et al., 2012). Here, SST shows a significant positive
correlation with observed IO (R = 0.52, P = 0.006, n = 48) south of the polar front above the
99 % confidence limit, but there is no correlation north of the polar front and only a weak
negative correlation using the combined dataset from the three campaigns (R = -0.18, P = 0.13,
n = 119).
Despite the above-mentioned point regarding the increase in observed IO levels in regions of
elevated chl-*a*, there is only a weak and negative correlation of IO with chl-*a* (both from
observations and satellite data) south of the polar front. However, there is a strong positive
relationship north of the polar front (R = 0.696, P = $2.3 \times 10^{-4}$, n = 29). In fact, for the region
north of the polar front, chl-*a* shows a significant positive correlation with observed IO above
the 99 % confidence limit (P < 0.001). The GEOS-Chem and CAM-Chem output also shows a





significant positive correlation (Fig. 8)which may result from the dependency of organic iodine
species on oceanic chl-*a* in both GEOS-Chem and CAM-Chem. Figure 8 shows a large
difference in correlation values for chl-*a* data obtained from observations and satellite (MODIS
Aqua, NASA, GSFC; https://oceancolor.gsfc.nasa.gov). In situ, observed chl-*a* showed an
improved correlation with IO compared to those with satellite chl-*a*. Figure 9 shows linear fits
for chl-*a* from in situ observations and satellite against IO for the entire dataset and north of
polar front subset. For the entire dataset, correlation of chl-*a* with IO from both observations
and satellite data is not significant. Chl-*a* from in situ observations positively correlates with
IO (R = 0.15, P = 0.32) while chl-*a* from satellite data correlates negatively (R = -0.13, P =
0.26). Correlations of chl-*a* with IO improves for the north of polar front for chl-*a* from
observations (R = 0.696, P = 0.0002), but chl-*a* from satellite data shows a statistically
insignificant correlation with IO (R = 0.08, P = 0.57). The discrepancies in chl-*a* from
observations and satellite data will make it difficult to identify links between the organic
parameter and atmospheric IO and expand this to a global scale.
Despite the observed negative relationship of IO with wind speed noted above, note that the
GEOS-Chem IO model output (which is dependent on the calculated HOI and $I_2$ fluxes) shows
a significant positive correlation with observed IO above the 99 % confidence limit for data
south (R = 0.78, P = P<0.001, n = 48) and north (R = 0.69, P = P<0.001, n = 68) of the polar
front, although there is no correlation across the entire dataset. Note that the model
underestimates IO values by 1 pptv south of the polar front and generally overestimates IO, by
~1.5 pptv, north of the polar front (Fig. 4). A linear fit for observed IO against modelled IO for
north and south of the polar front (Fig. 10) shows significant positive correlation of GEOS-
Chem output with observed IO, but with very different slopes north of the polar front (where
the models overestimate IO) and south of the polar front (where the models underestimate IO).





Hence, even though the correlations are good in the individual regions, the model does not
accurately reproduce the observed absolute concentrations.

**5. Conclusions**

In this study, region-specific parameterisation tools were devised for sea surface iodide (SSI)
estimation following previous SSI estimation methods from Chance et al. (2014) and
MacDonald et al. (2014). New observations of SSI from ISOE-9, SK-333 and BoBBLE (Indian
and the Southern Ocean) were used to create region-specific SSI parameterisations. An average
difference of up to 40 % in SSI concentration was observed among the existing
parameterisations (Eq. 1, 4, and 6) and the difference was 21 % for the region-specific ones
(Eq. 2, 3, and 5). Comparison of estimated SSI concentrations from various parameterisations
with observed SSI and sensitivity to seasonal salinity changes showed that the modified Chance
parameterisation (Eq. 2) was most suitable relative to the SST based parameterisation (Eq. 5)
for SSI estimation in the Indian Ocean and Southern Ocean region. Since the existing global
parameterisation schemes (Eq. 1 and 3) fail to match measured SSI in this region, it highlights
the need to conduct more observations of SSI in the Indian Ocean and Southern Ocean region
to fully understand and estimate the impact of seasonally varying, region-specific parameters
(like salinity, reversing winds patterns) influencing the seawater iodide concentration in this
region. Alternatively, a region-specific parameterisation scheme may be included in the global
models for better representation of seawater iodine chemistry in the Indian and Southern Ocean
region. Modelled estimates from Sherwen et al. (2019) also captured SSI well, although some
high concentrations in the northern Indian Ocean region were not captured. SSI estimation from
SST alone under-predicts SSI for the Indian Ocean, and so is not considered to be suitable for
SSI estimation in the Indian Ocean region. Although, improving SSI concentration in models
for the Indian Ocean and Southern Ocean region may improve the estimation of seawater iodine
chemistry, it does not translate to estimating the atmospheric iodine chemistry in this region.



An accurate estimation of inorganic iodine fluxes (HOI and $I_2$) is hence necessary to explain
observed levels of IO in the remote open ocean marine boundary layer. However, these first
concomitant observations of SSI and IO show that these inorganic fluxes, estimated in this
study, fail to explain detected IO levels for the entire dataset. No significant correlation was
seen between the SSI from different parameterisation techniques or estimated inorganic iodine
fluxes with observed IO levels. Fluxes estimated using iodide from different parameterisation
and measured iodide did not show large variation in values and followed a similar latitudinal
trend. This is indicative that the inorganic iodine flux parameterisation is not highly sensitive
to the SSI parameterisation. Predicted inorganic iodine fluxes did not explain iodine chemistry,
as indicated by IO levels, in the atmosphere above the Indian and Southern Ocean (Indian
Ocean sector). Chl-*a* shows a positive correlation with IO for the north of the polar front region,
suggesting that biologically emitted species could also play a role in addition to ozone and
iodide derived inorganic emissions of HOI and $I_2$. Finally, model predictions of IO
underestimate IO levels for the Southern Ocean region but overestimate IO in the Indian Ocean.
Models greatly underestimate IO in regions with higher chl-*a* concentration which could be
indicative of organic species playing a role (close to the Kerguelen Islands, refer Sect. 3.4.2).
This study suggests that the fluxes of iodine in the MBL are more complex than considered at
present and further studies are necessary in order to parameterise accurate inorganic and
organic fluxes that can be used in models. Using seawater iodide measurements and
calculations from different parameterisations did not alter the inorganic iodide flux estimate
greatly. Direct observations of HOI and $I_2$, alongside volatile organic iodine measurements in
the MBL are necessary in order to reduce the uncertainty in the impacts of iodine chemistry.
**6. Author contributions:**
ASM conceptualised the research plan and methodology. SI did the data curation, analysis, and
writing of the original draft. LT and RC did the iodide measurements provided unpublished





iodide data from ISOE-9, SK-333 and BoBBLE. PS and RCo provided salinity data for ISOE-
9. SCT and AUK provided chl-a data for ISOE-9. AKS and PVB provided chl-a data for SK-
333. AS and RR provided chl-a data from BoBBLE. CC and ASL did the CAM-Chem model
run for ISOE-9 and IIOE-2. TS did the GEOS-Chem model run for ISOE-9, IIOE-2 and ISOE-

8.

**7. Acknowledgements**
The authors thank the Ministry of Earth Sciences for funding the expeditions and IITM for
providing research fellowship to Swaleha Inamdar. We would particularly like to thank the
ISOE and IIOE-2 teams for their tireless contribution in manually recording and compiling
atmospheric and oceanic observations during the expedition. We express gratitude towards the
officers, crew and scientist on board RV S. A. Agulhas and RV Sagar Kanya ships for their
support. LJC, LT, RC and TS thank the UK NERC (NE/N009983/1) for funding.

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

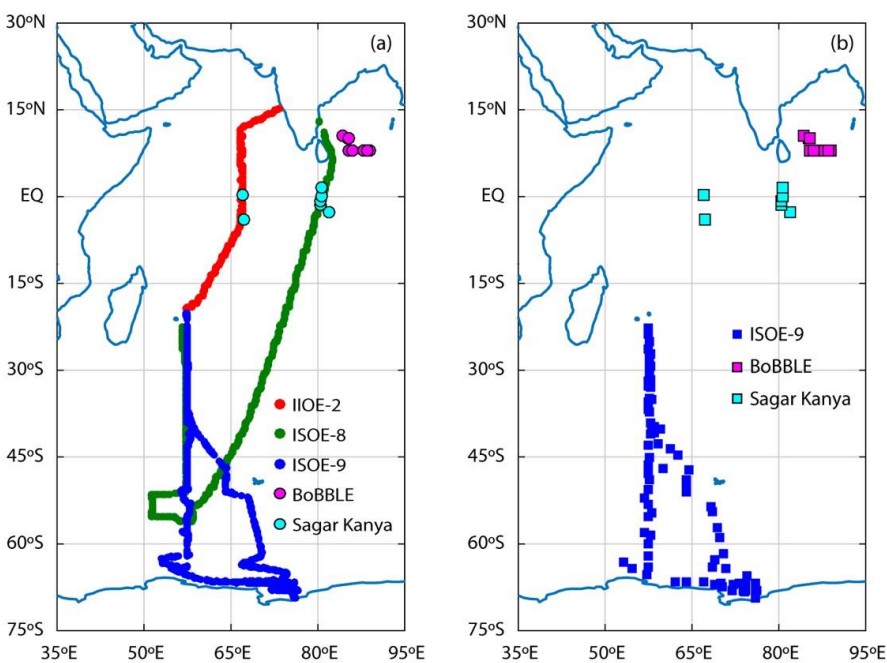


**Figure 1: Map of the Indian Ocean and the Southern Ocean (a) with cruise tracks for**
**campaigns conducted during the austral summer of 2014-2016. Green circles indicate the**
**cruise track for ISOE-8, red circles show the cruise track for IIOE-2, and blue circles**
**indicate the cruise track for ISOE-9. Magenta and cyan circles indicate sample locations**
**for the BoBBLE and SK-333 expeditions respectively. (b) boxes represent 129 seawater**
**iodide sampling locations from 3 expeditions following the colour code in (a).**





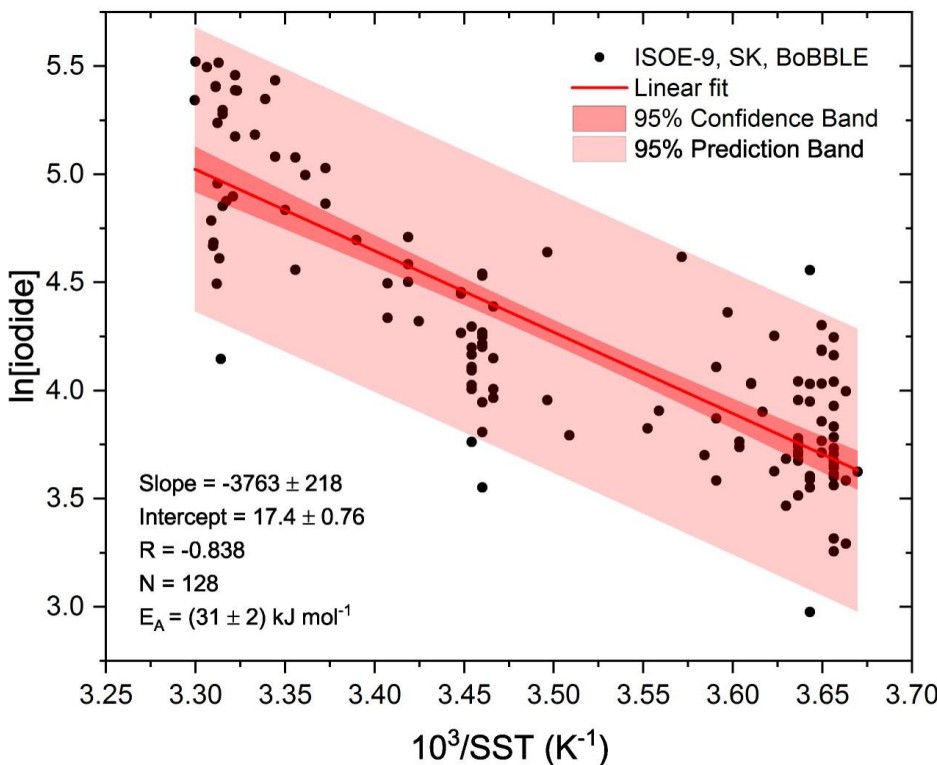


**Figure 2: Arrhenius form plot of sea surface iodide concentrations against SST from all available seawater iodide field observations in the Indian Ocean and Southern Ocean. The red line represents a linear fit., the shaded region in dark red (inner) indicates the 95% confidence bands and shaded area in light red (outer) indicates the 95% prediction bands.**









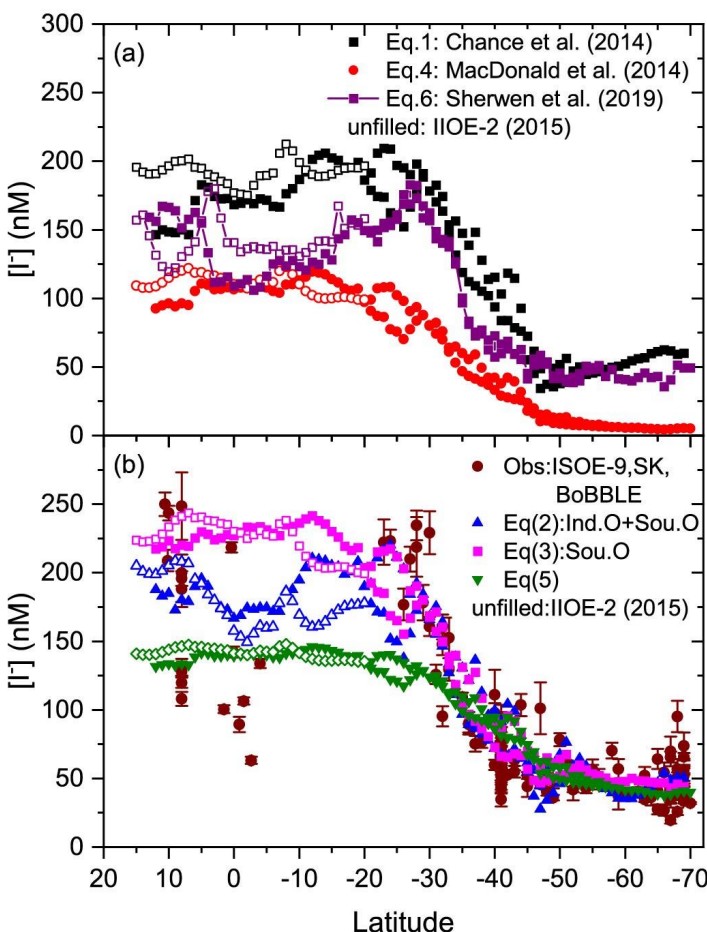


**Figure 3: Latitudinal averages of calculated sea surface iodide (SSI) concentrations for each campaign using (a) existing, (b) new parameterisation tools and observations from ISOE-9, SK-333, and BoBBLE. Filled markers represent combined SSI from ISOE-8 and ISOE-9, unfilled markers represent SSI from IIOE-2 campaign.**

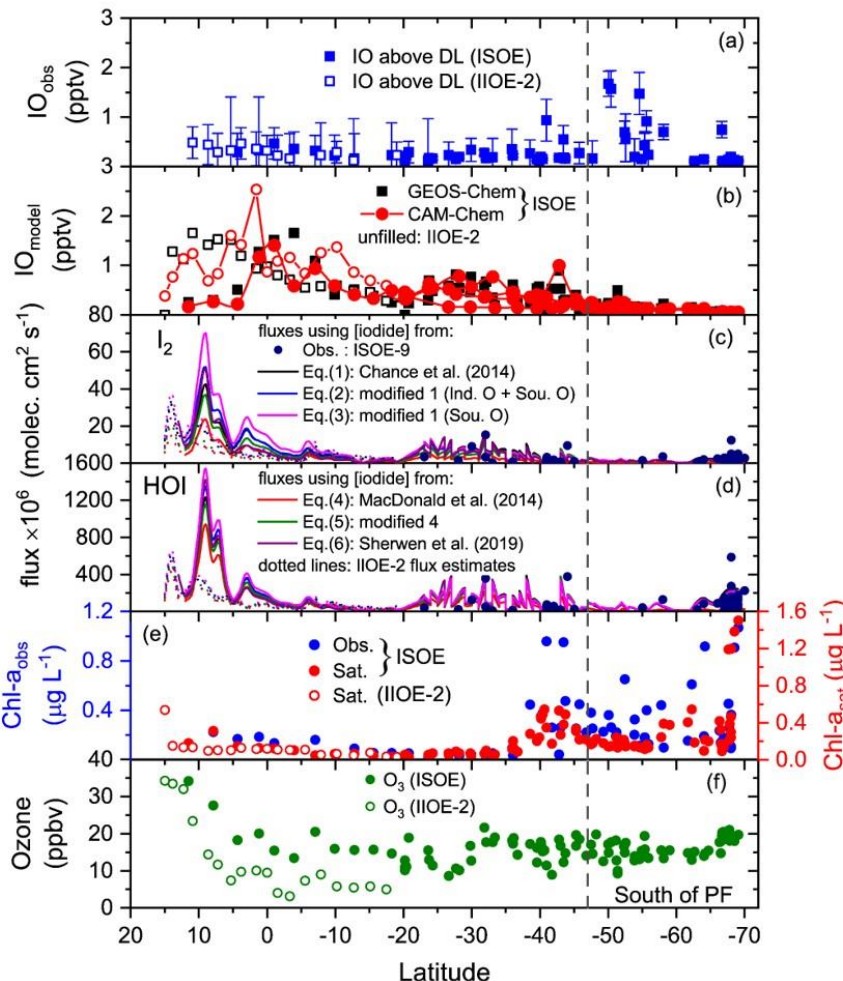

**Figure 4: Daily averaged atmospheric and oceanic parameters combined from ISOE-8, IIOE-2, and ISOE-9 field campaigns. Data marked ISOE represents combined data from ISOE-8 and ISOE-9. Unfilled markers and dotted lines show values for IIOE-2. (a) IO above detection limit from ISOE-8, ISOE-9 and IIOE-2. (b) Surface IO values from GEOS-Chem and CAM-Chem models. (c) and (d) comprise of HOI and I₂ fluxes estimated from Eq. (7) and (6) respectively. Fluxes are colour coded for different sea surface iodide (SSI) datasets used for their estimation. Colours black, blue, red and green correspond to fluxes calculated using SSI estimation from Eq. (1) to (5), purple colour represents the use of model SSI predictions** (Sherwen et al., 2019b)**, filled circles in dark blue correspond to measured SSI from ISOE-9 for each observation, (e) chlorophyll-*a* observations from ISOE-8 and ISOE-9 (blue circles) and satellite data for all campaigns (red circles). (f) ozone mixing ratios from campaigns ISOE and IIOE-2. The dashed line**



**marks the polar front at 47° S. Observational plots for ISOE-8 and IIOE-2 were adapted**
**from Mahajan et al. 2019 a & b.**

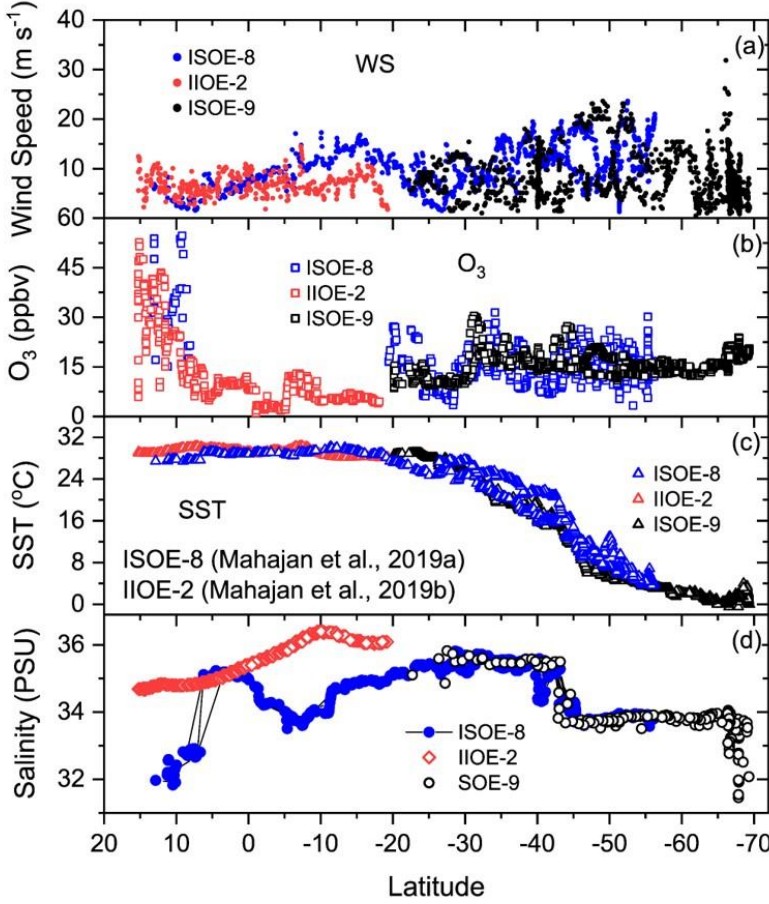


**Figure 5: Latitudinal plot of hourly-averaged field measurements of wind speed, ozone**
**mixing ratios, SST and salinity[†] from ISOE-8, IIOE-2, and ISOE-9 campaigns. Data**
**markers in red belong to the IIOE-2 campaign; those in blue belong to the ISOE-8 and**
**markers in black are from ISOE-9 for all the panels. Observational plots for ISOE-8 and**
**IIOE-2 were adapted from Mahajan et al. 2019 a &b.**

[†] Salinity data for IIOE-2 are monthly climatological means from World Ocean Atlas as described in the supplementary text.



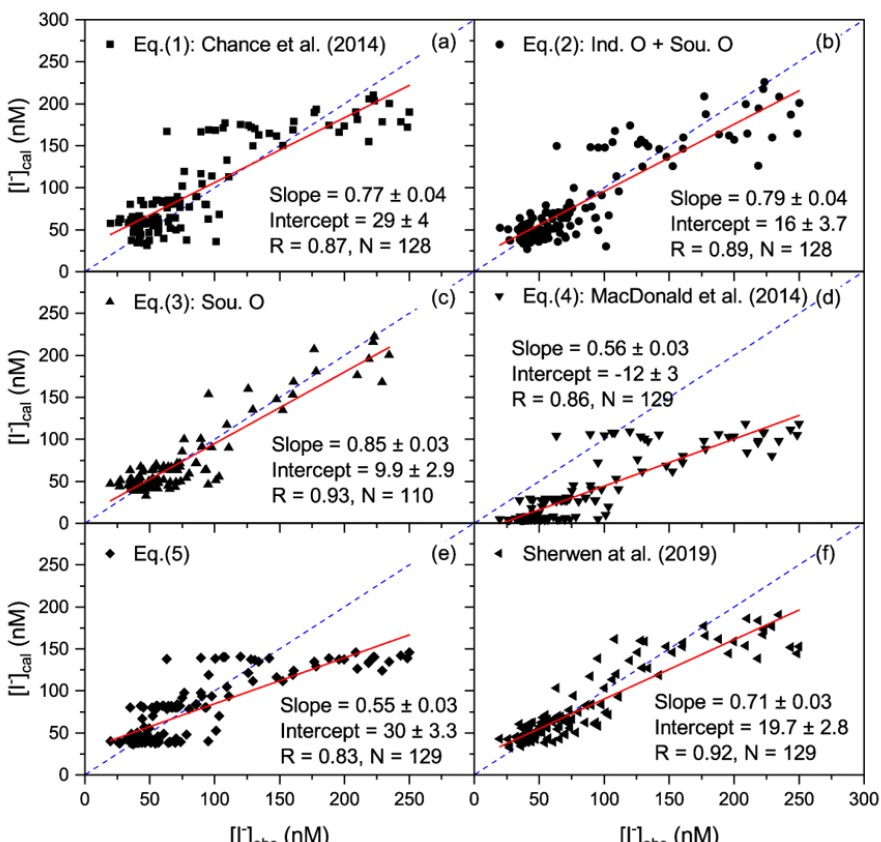

**Figure 6: Linear fit analysis of estimated sea surface iodide (SSI) concentrations (y axis) from parameterisation methods in Eq. (1) to (5) and model prediction (Sherwen et al., 2019) against measured SSI concentration (x axis) from ISOE-9, SK-333 and BoBBLE. In panel (c) SSI are compared only with ISOE-9 observations for Southern Ocean specific parameterisation. R represents Pearson's correlation coefficient and N is the size of the dataset. Dashed blue line represents identity (1:1) line.**

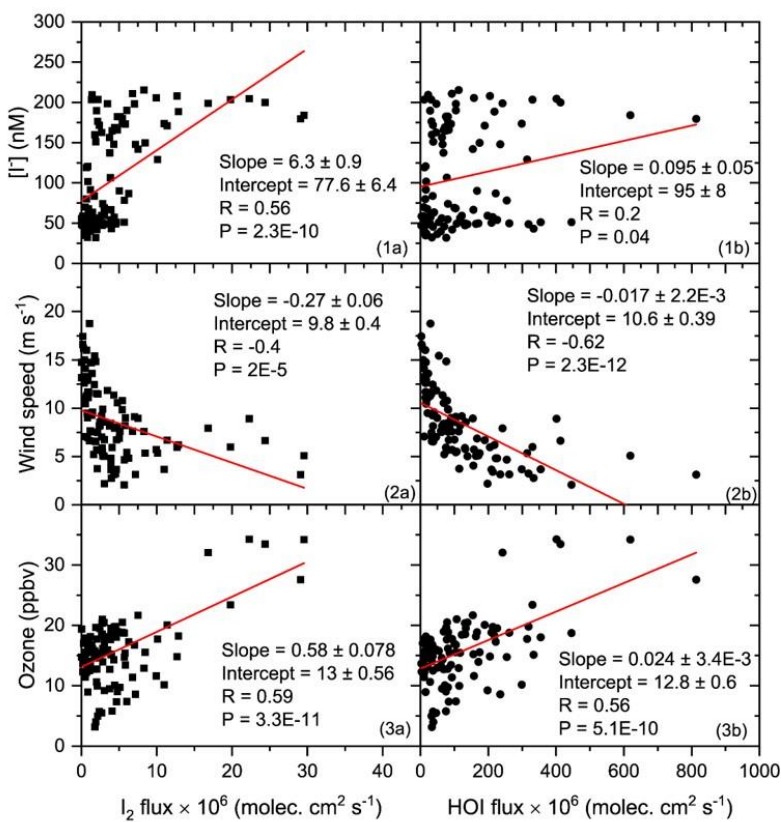

**Figure 7: Linear fit of daily average sea surface iodide (SSI) concentration, wind speed**

**and ozone mixing ratio (y axis) against calculated I₂ and HOI flux (x axis) against for all**

**the campaign. HOI and I₂ are calculated using SSI estimated using the modified Chance**

**parameterisation for Indian Ocean and Southern Ocean in Eq. (2).**




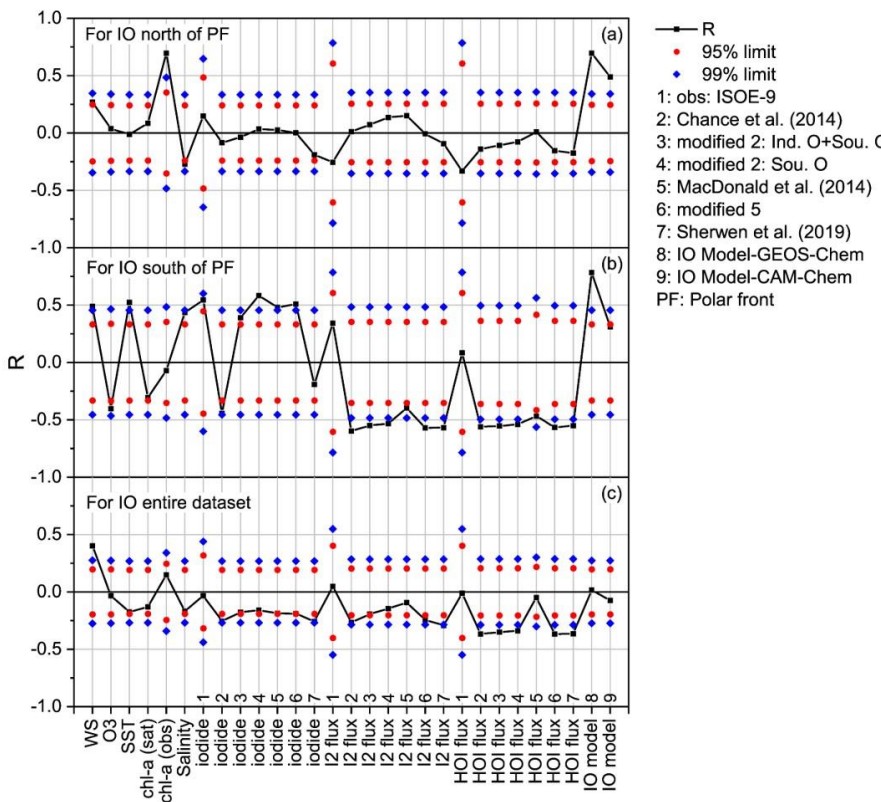


**Figure 8: Pearson's correlation coefficient of observed iodine monoxide (IO) with oceanic**
**and atmospheric parameters combined for ISOE-8, IIOE-2, and ISOE-9 campaigns.**
**Correlations are performed for daily averages of IO and corresponding parameters listed**
**on the X axis. The black squares represent Pearson's correlation coefficients (R), the**
**diamonds (blue) mark the 99% confidence limit, and the circles (red) correspond to the**
**95% confidence limits in all the panels, (a) includes data from all campaigns to the north**
**of the polar front (n = 72), (b) represents combined data for the south of the polar front**
**(n = 48), the last panel (c) includes the entire dataset from three campaigns (n = 120).**



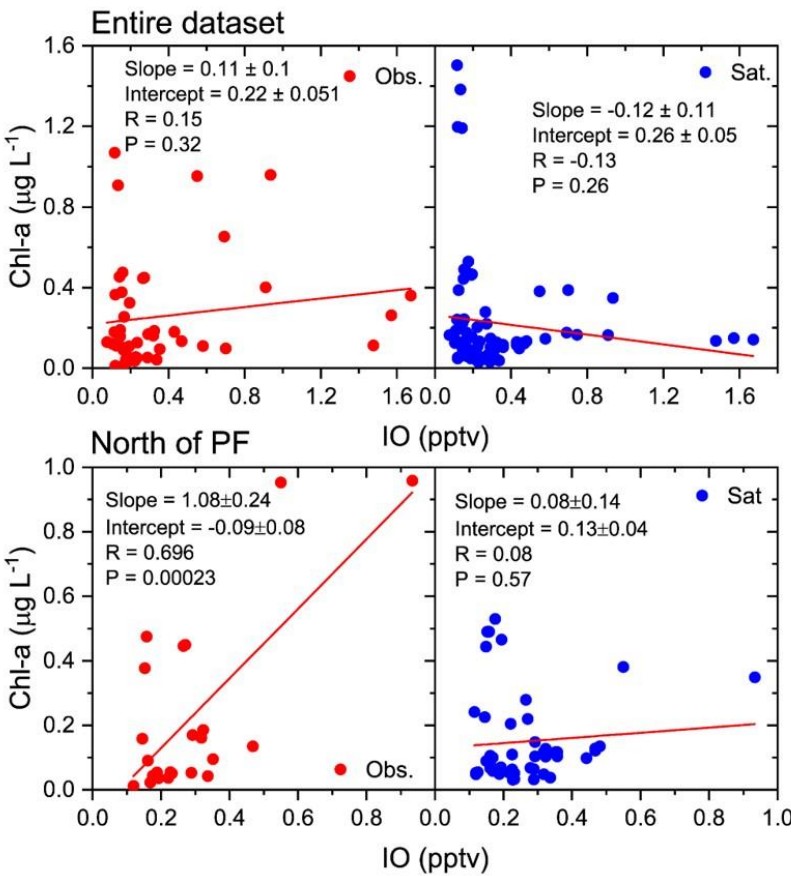


**Figure 9: Linear fit of daily averaged field observations of chlorophyll-*a* (red circles) and chlorophyll-*a* satellite data (blue circles) (y axis) against observed iodine monoxide (IO) (x axis) from ISOE-8, IIOE-2, and ISOE-9 campaigns. The top panel includes chlorophyll-*a* for the entire dataset; the bottom panel includes data to the north of the polar front.**


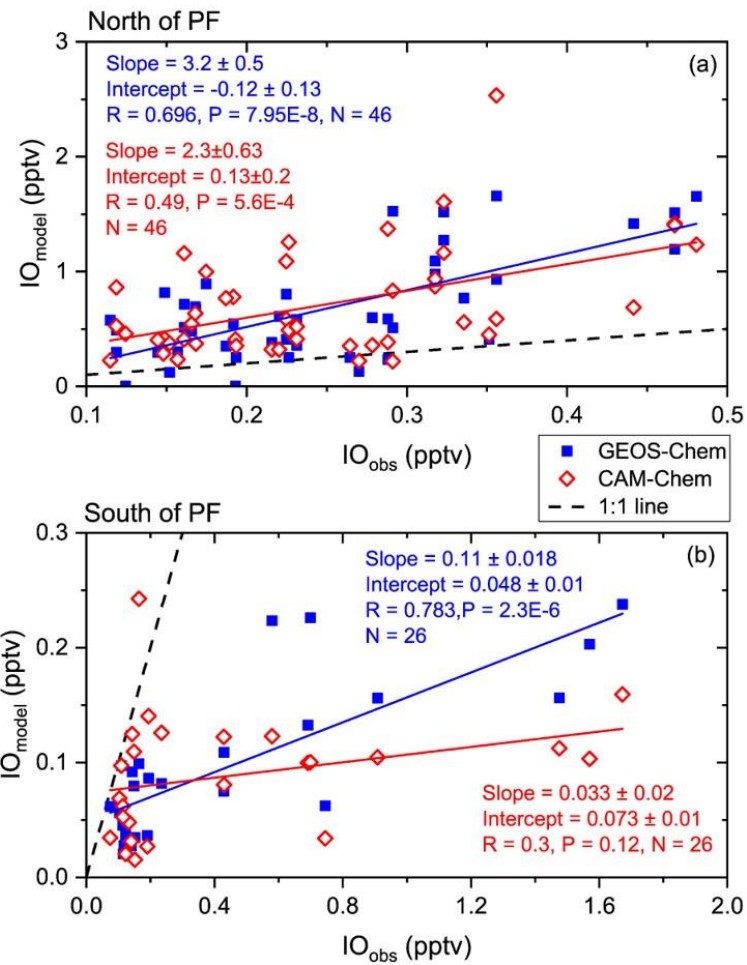


**Figure 10: Linear fit of daily averages of modelled surface iodine monoxide (IO) output**
**(y axis) from GEOS-Chem (filled blue squares) and CAM-Chem (unfilled red diamonds)**
**against observed IO (x axis) for ISOE-8, IIOE-2 and ISOE-9 campaigns. (a) includes**
**linear fits of both GEOS-Chem and CAM-Chem for IO detected to the north of the polar**
**front, (b) shows the same for the region south of the polar front. Two data points in panel**
**(a) at 41º S and 43º S are removed due to large differences between observation and**
**modelled values.**








**10. Tables**

| Expedition | Research Vessel | Duration | Location | Meridional Transect | Observations |
|---|---|---|---|---|---|
| 8th Indian Southern Ocean Expedition (ISOE-8) | Sagar Nidhi, India | 7 Jan 2015 to 22 Feb 2015 | Indian Ocean from Chennai, India to Port Louis, Mauritius | 13º N to 56º S | IO, O$_3$ |
| 2nd International Indian Ocean Expedition (IIOE-2) | Sagar Nidhi, India | 4 to 22 Dec 2015 | Indian Ocean from Goa, India to Port Louis, Mauritius | 15º N to 20º S | IO, O$_3$ |
| Bay of Bengal Boundary Layer Experiment (BoBBLE) | R.V. Sindhu Sadhana | 23 June 2016 to 24 July 2016 | Southern Bay of Bengal | 8º N to 10º N | Seawater samples for I$^-$ |
| Sagar Kanya-333 (SK-333) | Sagar Kanya, India | 5 Sept 2016 to 20 Sept 2016 | Southern Arabian Sea and Southern Bay of Bengal | 1.6º N to 4º S | Seawater samples for I$^-$ |
| 9th Indian Southern Ocean Expedition (ISOE-9) | S A Agulhas, South Africa | 6 Jan 2017 to 26 Feb 2017 | Indian and Southern Ocean from Port Louis, Mauritius to Antarctica | 20º S to 70º S | IO, O$_3$, I$^-$ |


**Table 1: Details of the three expeditions contributing to the IO and seawater iodide**
**dataset in this study. Expeditions are listed in chronological order from 2014 to 2016.**








| Eq. No | Reference | Parametric equation ([iodide] in nM) | Database location | Data points | $R^{2*}$ | $R^2$ |
|---|---|---|---|---|---|---|
| *Eq. 1* | *Chance et al. (2014)* | $[iodide] = 0.28(\pm0.002) \times sst^2 + 1.7(\pm0.2) \times |latitude| + 0.9(\pm0.4) \times [NO_3^-] - 0.02(\pm0.002) \times MLD_{pt} + 7(\pm2) \times salinity - 309(\pm75)$ | *Majorly Atlantic and Pacific Ocean* | *n = 673* | *0.676* | *0.758* |
| Eq. 2 | This study | $[iodide] = 0.36(\pm0.04) \times sst^2 - 2.7(\pm0.5) \times |latitude| + 0.28(\pm0.57) \times [NO_3^-] + 0.64(\pm0.17) \times MLD_{pt} - 5.4(\pm3.82) \times salinity + 22(\pm137)$ | Indian and Southern Ocean | n = 128 | 0.794 | 0.794^ |
| Eq. 3 | This study | $[iodide] = 0.25(\pm0.017) \times sst^2 - 0.6(\pm0.4) \times |latitude| + 2.2(\pm0.4) \times [NO_3^-] - 5.5(\pm3.3) \times salinity + 212(\pm123)$ | Southern Ocean | n = 110 | 0.859 | 0.859^ |
| *Eq. 4* | *MacDonald et al. (2014)* | $[iodide] = 1.46 \times 10^{15} \times exp\left(\frac{-9134}{SST}\right)$ | *Atlantic, Central and West Pacific Ocean* | *n = ~88* | *0.71* | *0.739* |
| Eq. 5 | This study | $[iodide] = 3.6 \times 10^7 \times exp\left(\frac{-3763}{SST}\right)$ | Indian and Southern Ocean | n = 129 | 0.702 | 0.697^ |
| *Eq. 6* | *Sherwen et al. (2019)* | *Machine learning based regression approach* | *Atlantic, Pacific, Indian and Southern Oceans* | *n = 1293* | - | *0.842* |


**Table 2: List of existing global (italics) and new region-specific (regular) parameterisations for sea surface iodide concentration indicating data location and number of data points used to formulate each equation. Here [iodide] represents sea surface iodide concentration in nM, sea surface temperature as 'sst' in ºC, and SST in K. Nitrate concentration ($[NO_3^-]$) is given in µmol L$^{-1}$, mixed layer depth as $MLD_{pt}$ in m, subscript 'pt' indicates potential temperature implying a temperature change of 0.5 ºC from the ocean surface (Monterey and Levitus, 1997), and salinity in PSU. Further details on individual parameters and the choice of Eq. (1) over others proposed in Chance et al. (2014) are discussed further in the supplementary text. $R^{2*}$ represents the initial coefficient of determination (COD) while deriving each parameterisation, and $R^2$ represents COD from correlation analysis of the calculated iodide with observations in this study (ISOE-9, SK-333, BoBBLE).**





$^\wedge$**Higher $R^2$ values for the modified parameterisations reflect the fact that they have been**

**derived using the same observational data as they are tested on.**