# Peer review of "Estimation of Reactive Inorganic Iodine Fluxes in the Indian and Southern Ocean Marine"

_Atmospheric Chemistry and Physics, 2019_

## Referee Comment (RC1) · Anonymous Referee #1 · 8 Apr 2020

The paper uses concurrent observations of sea surface iodide (SSI), ozone (O3), and iodine monoxide (IO) along with several other parameters to assess different methods of estimating iodine fluxes to the marine boundary layer. Region-specific forms of these methods for the Indian Ocean and Southern Ocean are derived and further assessed against the observations. The results are substantially different in the Indian Ocean and Southern Ocean and on either side of the polar front. Furthermore, the results are often contrary to previous findings, highlighting the need for further studies. The fundamental finding of the paper that existing methods fail to reproduce observations and that consistent improvements applicable to the full data set are not forthcoming is worthy of publication; however, the authors must better demonstrate and communicate

this with robust statistical tests. Toward this point I have the following specific major comments:

1) As the title states the observational region encompasses the Indian Ocean and Southern Ocean, however, analysis of (SSI) in the Indian Ocean alone fails to obtain a significant result. As the authors state this is likely due to the limited statistics (N=18). This finding calls into question whether the application of the combined fitted result should be applicable to the Indian Ocean. ANOVA or similar methods should be applied to determine whether the Indian Ocean is statistically different. In particular, an F-test should be conducted. The presented results suggest that spatial and temporal differences between the measurement campaigns, and other effects may present confounding variables to such methods. Despite this, even "failed" statistical tests with inconclusive results are needed for proper framing of the results in the Indian Ocean.

2) Similar statistical analysis is also needed for the different analyses north and south of the polar front. Pearson coefficients for the correlation of observed IO with various parameters divided and combined data set are shown in Fig. 8. These help give some idea of the differences between the correlations on either side of the front, but the picture is incomplete. Two particular results highlighted in the text are demonstrative: GEOS-Chem modeled IO is significantly correlated with observed IO in the data subsets but not in the complete data set. Fig. 8 shows this also the case for CAM-Chem at ∼94% confidence as well. The reason for this, as stated by the authors, is that the variability in both models across the polar front is significantly different than observed. Further, the reader can see this for themselves in Figs. 4 and 10. In contrast, correlations with chl-a are much more difficult to interpret. Most data occupy a limited dynamic range in Fig. 4 and the correlation plots in Fig. 9 indicate individual points may be driving the correlation but this is not clear. In both instances systematic statistical assessment of the data subsets would be helpful.

In addition I have the following additional two major comments. 3) As the authors state in the abstract their results start from "the first concomitant observations of iodine oxide

(IO), O3 in the gas phase, and sea surface iodide concentrations." The choice of "concomitant" implies some intrinsic link between the set of parameters, and theoretically these parameters are expected to correlate via the mechanism outlined by Eqs. 7 and 8. However, the authors ultimately find that the correlations are the opposite of those in the equations as shown in Fig. 8 and discussed in the text. Keeping close to the underlying data these correlations should be shown in full similar to Chl-a in Fig. 9.

4) The underlying measurements are contained in four other papers (Chance et al., 2019a,b and Mahajan et al., 2019 a,b) are fundamental to assessing these findings. These are sufficiently critical to the results presented that I would recommend linking them as companion papers. I cannot locate Chance et al. (2019b) and do not believe it is published. The measurements are described in part in Chance et al. (2019a) nonetheless it is troubling that such critical measurements are neither published nor described more fully. If the measurements are not yet published the description in this paper should be expanded. Notably, MAX-DOAS data (which are published elsewhere) appear in the supplement while the SSI data do not.

I have following minor and technical comments. Line 28: As commented above, "concomitant" here may be misleading in the abstract. While the latter portion of the abstract (L 37-39) makes clear that "Sea-air fluxes ... calculated from the atmospheric ozone and seawater iodide ... failed to adequately explain the detected IO in this region" it does not make clear that the correlations are largely null or even contrary to expectations. Line 58-60: Saiz-Lopez et al. (2006b) included IO condensation onto particles in order to explain particle growth. However, I do not believe there is any claim of direct nucleation by IO. This sentence should be clarified. Line 73 and 74: Why are these reactions not labeled and numbered? Line 266: The detection limit and precision should match units to be more easily compared. Line 270-272: Consider simplifying description of wind flagging to inclusion of the forward hemisphere or the like. Presently mildly confusing. Line 302-30: Wind speed is first introduced here but discussed frequently hereafter. What is the wind speed referred to? e.g. is it U10 or some

other standard? This is particularly relevant later when comparing with model treatment of wind parameters. Line 305 - 307: As is made clear later on line 317, neither of the activation energies determined in MacDonald et al. (2014) is significantly different from zero. While it does not examine the products separately, Magi et al. (1997) does find a significant activation energy for the first I- + O3 reaction. MacDonald et al. noted that the Arrhenius pre-exponential factor in Magi et al. is ten orders of magnitude greater than the diffusion limit to justify assuming an activation energy for the first step of zero. More recent work has also called the rates determined in Magi et al. into question (Moreno et al., 2018), however, the confounding factors (high iodide concentrations and iodide surface activity) cannot fundamentally dispel the observed positive temperature dependence. Notably, the values reported in MacDonald are predicated on an assumption that the activation energy of I- + O3 is zero and it is even more uncertain whether the overall activation energy to produce I2 is negative. The activation energies from MacDonald are better summarized as approximately zero (e.g. Moreno and Baeza-Romero 2019) as the overall temperature dependence remains unresolved. Line 401-405: While diurnal variation in O3 can be inferred, its reversal is not apparent in Fig. 5b as referred to. Line 645 - 646: How do the authors infer that photochemistry is responsible for the differences? This is not obvious to me. Line 654 – 655: From Fig. 7 it appears that the p-value for the HOI correlation is 0.04. Given that 5% significance is used as a standard elsewhere in the paper this would indicate that HOI does show significant correlation contrary to this statement. Line 771: VOI not previously introduced. Figure 3: The literature calculations are not readily compared with the observations as they are in separate panels. The observations should appear in both panels. Figure 9: It should be made clear that the IO here is observed as modeled IO is also presented elsewhere. The legends readily blend in with the scattered data and should be made more clearly separate. Table 2: The database location column should be moved left of the equations as it is otherwise unclear in isolation what the differences between Eq. 2, 3, and 5 are. Iodide and nitrate concentrations should be given with consistent (though not necessarily the same) units, i.e. M or mol L-1 but not

both. Table 2 and Supplement: In addition to the expectation of higher R2 values for fitting to the same data set, equations having more degrees of freedom are expected to have better fits. The adjusted R2 values should be used for proper comparison of the overall parameterizations. If this is already the case it should be described as such in the supplement.

References: Chance, R., Tinel, L., Sherwen, T., Baker, A., Bell, T., Brindle, J., Campos, M. L. A. M., Croot, P., Ducklow, H., He, P., Hoogakker, B., Hopkins, F. E., Hughes, C., Jickells, T., Loades, D., Macaya, D. A., Mahajan, A. S., Malin, G., Phillips, D. P., Sinha, A. K., Sarkar, A., Roberts, I. J., Roy, R., Song, X., Winklebauer, H. A., Wuttig, K., Yang, M., Zhou, P. and Carpenter, L. J.: Global sea-surface iodide observations, 1967-2018, submitted, doi:10.5285/7e77d6b9-83fb-41e0-e053-6c86abc069d0, 2019a. Chance, R., Tinel, L., Carpenter, L. J., Sarkar, A., Sinha, A. K., Mahajan, A. S., Chacko, R., Sabu, P., Roy, R., Jickells, T. D., Stevens, D. and Wadley, M.: Surface inorganic iodine speciation in the Indian Ocean and Indian Ocean sector of the Southern Ocean, Manuscr. Prep., 2019b. MacDonald, S. M., Gómez Martín, J. C., Chance, R., Warriner, S., Saiz-Lopez, A., Carpenter, L. J. and Plane, J. M. C.: A laboratory characterisation of inorganic iodine emissions from the sea surface: Dependence on oceanic variables and parameterisation for global modelling, Atmos. Chem. Phys., 14(11), 5841–5852, doi:10.5194/acp-14-5841-2014, 2014. Magi, L., Schweitzer, F., Pallares, C., Cherif, S., Mirabel, P. and George, C.: Investigation of the Uptake Rate of Ozone and Methyl Hydroperoxide by Water Surfaces, J. Phys. Chem. A, 101(27), 4943–4949, doi:10.1021/JP970646M, 1997. Mahajan, A. S., Tinel, L., Hulswar, S., Cuevas, C. A., Wang, S., Ghude, S., Naik, R. K., Mishra, R. K., Sabu, P., Sarkar, A., Anilkumar, N. and Saiz Lopez, A.: Observations of iodine oxide in the Indian Ocean Marine Boundary Layer: a transect from the tropics to the high latitudes, Atmos. Environ. X, 1(January), 100016, doi:10.1016/j.aeaoa.2019.100016, 2019a Mahajan, A. S., Tinel, L., Sarkar, A., Chance, R., Carpenter, L. J., Hulswar, S., Mali, P., Prakash, S. and Vinayachandran, P. N.: Understanding Iodine Chemistry Over the Northern and Equatorial Indian Ocean, J. Geophys. Res. Atmos., (x), 2018JD029063, doi:10.1029/2018JD029063,

2019b. Moreno, C. G., Gálvez, O., López-Arza Moreno, V., Espildora-García, E. M. and Baeza-Romero, M. T.: A revisit of the interaction of gaseous ozone with aqueous iodide. Estimating the contributions of the surface and bulk reactions, Phys. Chem. Chem. Phys., 20(43), 27571–27584, doi:10.1039/c8cp04394a, 2018. Moreno, C. and Baeza-Romero, M. T.: A kinetic model for ozone uptake by solutions and aqueous particles containing I- and Br-, including seawater and sea-salt aerosol, Phys. Chem. Chem. Phys., 21(36), 19835–19856, doi:10.1039/c9cp03430g, 2019

Please also note the supplement to this comment:
https://www.atmos-chem-phys-discuss.net/acp-2019-1052/acp-2019-1052-RC1-supplement.pdf

---

## Referee Comment (RC2) · Anonymous Referee #2 · 9 Apr 2020

The paper by Swaleha Inamdar et al shows new and simultaneous measurements of iodine oxide (IO), ozone (O3) in the gas phase, and sea surface iodide (I-; SSI) concentrations during the ISOE-9 ship campaign in the Indian Ocean and Southern Ocean in January-February 2017. These measurements are complemented with previously published ship based measurements in the Indian Ocean and Bay of Bengal and with different available parametrizations to compute the iodine (I2) and hypoiodous acid (HOI) fluxes. This study includes important new results which should be publishable after a detailed and careful major revision of the manuscript taking all comments into account.

[Figure]

General comments:

Earlier studies: The paper misses to refer to other iodine ship based studies, such as Hepach et al (2016), where iodocarbons, IO, and many different biological parameters were observed and possible biological production mechanisms were discussed. A positive correlation between iodine sources and biology and a biology control is not a new result. This has to be taking into account in the abstract, introduction, discussion, and conclusions of your results. There were earlier ship based measurements of atmospheric IO, I2 and ozone, f.e., in the tropical West Pacific during e.g. the SHIVA campaign (Pfeilsticker et al 2013) and in the Indian Ocean with the OASIS campaign (Krüger et al 2015) which should be mentioned and related to.

Indian Ocean: What about the strong seasonality of the Indian Ocean, physical and biological, which may play an important role for the interpretation of your results? This needs to be included in the introduction and the discussion (see Schott and McCreary 2001; SIBER Report No. 1, 2011). Your paper should go beyond a correlation based only discussion. What are the mechanisms in the Indian Ocean: Any biology, ocean and atmospheric circulation impacts? It would be very interesting to get some more details on the spatial distribution of your observed in-situ quantities compared to satellite and/or global model data, adding maps of e.g. SST, Chl-a, wind, SSI/I2.

Measurement, flux parametrization, and model details, errors and uncertainties:

What are the error bars of the measurements especially of SSI and what are the uncertainties of the flux estimates? This needs a careful and detailed discussion in the ms. The observed SSI (Chance et al 2019b under review) study is not available to the readership so that we cannot find any information about the kind of measurements nor the quality. What was the measurement strategy (day vs night time, how often etc)? Where were the surface water iodide measurements carried out onboard of the ship and when? Does the diurnal cycle play a role? Substantial measurement details are missing and need to be added to understand your ship measurement and study design

better.

For the observed meteorological data, surface wind is conventionally given as 10 min averages and then there are gusts (instantaneous wind). Currently, you use hourly averages which lead to a smoothing of the average wind speed and thus impact your flux parametrization calculations which are based on a threshold limit of 14 m/s. Next, at which altitude levels onboard of the ship were your wind and others quantities measured? Conventionally 10 m surface wind is used for flux calculations. What did you use and on what are the flux parameterisations based on? The measurement section, data and graphs need a thorough and detailed revision.

Substantial details are also missing for the flux parametrization. How well do the estimated iodine fluxes explain observed surface atmospheric I2 concentrations? What are the largest uncertainties also in contrast to the common bulk parametrizations of air-sea fluxes which have a very high (>50%) uncertainty especially with regard to the role of the wind ?

What are the main chemistry transport and chemistry climate model uncertainties? What is the role of the meteorology and ocean surface (composition and circulation); is this consistently taken into account in these two models compared to your observations?

Specific comments:

Line 127-129: Grossmann et al 2013 and others published remote open ocean data. Please rephrase the sentence.

Table 1: There are no 2014 measurements listed in the third column although you mention this in the table caption, abstract, introduction etc.

Technical corrections:

Figures and figure captions: The graphs and figure captions are not self-explaining and not presented in a consistent way. The acronyms are mostly not introduced nor are the

figures easy to relate to each other, f.e. ozone in Figure 4 and 5 is it the same? What is PF; I assume Polar Front and where is this in Fig 5? All your figures and figure captions need a thorough revision.

References:

Hepach, H. et al., Biogenic halocarbons from the Peruvian upwelling region as tropospheric halogen source, Atmos. Chem. Phys., 16, 12219–12237, 2016.

Krüger K. et al, OASIS-research cruises SO234-2 and SO235 of R/V SONNE in summer 2014 in the tropical Indian. . ., The Indian Ocean Bubble, Issue No., 3, Aug. 2015.

Pfeilsticker K. et al, The SHIVA Western Pacific Campaign in Fall 2011, Malaysian Journal of Science 32 (SCS Sp Issue), 141-148, 2013.

Schott, F.A. and McCreary, J.P., 2001. The monsoon circulation in the Indian Ocean. Progress In Oceanography, 51(1): 1-123.

SIBER Report No.1, Sustained Indian Ocean Biogeochemistry and Ecosystem Research, 2011. Research, 2011.

---

## Referee Comment (RC3) · Anonymous Referee #3 · 6 May 2020

The paper by S. Inamdar is using a large data set of seawater iodide, atmospheric ozone and atmospheric IO concentrations to test the reactive inorganic iodine fluxes calculated from different parameterisations of seawater iodide,. The authors propose new parameterisations of seawater iodide that are specific for given regions of the global ocean, and compared to already established parameterisation for the global ocean. They find that the parameterisation used has little impact on the computed atmospheric IO concentrations. Observed IO concentrations cannot be adequately computed using inorganic iodine fluxes and chemistry. As IO is correlated to Chl-a, the authors suggest a biogenic impact on iodine in the region investigated. The paper is well and clearly written and organized. Iodine fluxes, chemistry and impacts on the

atmospheric composition are poorly understood and this study brings a nice input into our understanding. I suggest the paper is published after only minor comments (below) are taken into account.

Minor comments

Section 2.1 iodide parameterisations

Lines 201 to 218 : the argumentation on the need to have regional parameterizations should go in the introduction ?

Line 226 : would be nice to recall why sea surface nitrate concentrations were chosen as a parameter influencing iodide concentrations

Section 2.2 ozone measurements

Contaminations on a ship may occur from other sources than the ship's smokestack (such as cooking exhausts, or air conditioning exhausts). Were there any indicator of anthropogenic compounds concentrations available to exclude contaminations?

3.Results

3.2 Iodide line 432-433: the end of the sentence is not clear, please reformulate 3.3 Iodine fluxes line 491: premature to mention discrepancies between modelled and measured IO in this section? Would better fit in the discussion section

4. Discussion

line 712: concerning the lack of correlation with satellite base Chl-a while in situ Chl-a concentrations are correlated to observed IO concentrations. May this be due to geographical differences in what biological species Chl-a represent in these different regions, or may be due to uncertainties in the Chl-a retrieval from satellite, or even also scaling problems. Did the authors try to extract satellite Chl-a where the actual Chl-a in situ measurements were performed to compare one with the other?

---

## Author Response (AR1)

**Response to reviewer comments for manuscript number: acp-2019-1052**

Comments by reviewers are shown in italic typeface and the responses shown normal typeface.

Referee comments:

Reviewer #1:

*The paper uses concurrent observations of sea surface iodide (SSI), ozone (O3), and iodine monoxide (IO) along with several other parameters to assess different methods of estimating iodine fluxes to the marine boundary layer. Region-specific forms of these methods for the Indian Ocean and Southern Ocean are derived and further assessed against the observations. The results are substantially different in the Indian Ocean and Southern Ocean and on either side of the polar front. Furthermore, the results are often contrary to previous findings, highlighting the need for further studies.*

*The fundamental finding of the paper that existing methods fail to reproduce observations and that consistent improvements applicable to the full data set are not forthcoming is worthy of publication; however, the authors must better demonstrate and communicate this with robust statistical tests. Toward this point I have the following specific major comments:*

RESPONSE: We thank the reviewer for providing detailed constructive comments and suggestions. The following is a point by point response to the review with corresponding changes made to the manuscript. We hope that the manuscript will now be acceptable with these changes.

*1) As the title states the observational region encompasses the Indian Ocean and Southern Ocean, however, analysis of (SSI) in the Indian Ocean alone fails to obtain a significant result. As the authors state this is likely due to the limited statistics (N=18).*
*This finding calls into question whether the application of the combined fitted result should be applicable to the Indian Ocean. ANOVA or similar methods should be applied to determine whether the Indian Ocean is statistically different. In particular, an F-test should be conducted. The presented results suggest that spatial and temporal differences between the measurement campaigns, and other effects may present confounding variables to such methods. Despite this, even "failed" statistical tests with inconclusive results are needed for proper framing of the results in the Indian Ocean.*

RESPONSE: As per the above suggestion, the inconclusive results for Indian Ocean region are now incorporated in the manuscript. Table S1 (supplementary text) is revised to include the results of initial linear regression analysis for individual parameters from the Indian Ocean region. This analysis is similar to that of Eq. (2) and (3) for Indian + Southern Ocean and Southern Ocean respectively. The values of coefficient of determination ($R^2$), slope, intercept and p (at 5%) indicate that for this region only the absolute latitude ($R^2 = 0.3$, $p = 0.02$) and salinity ($R^2 = 0.3$, $p = 0.02$) parameters show statistically significant dependence on the observed sea surface iodide (SSI) concentration. A parameterisation formulated using these parameters is listed in the manuscript Table 2 as Eq. (3a) for Indian Ocean. ANOVA test on dataset for Eq. (3a) provides F ratio of 3.604 and $p = 0.053$ which indicates that the null hypothesis is accepted as the F ratio is larger than the critical F value from an f-distribution table for (2,15) degrees of freedom. Similarly, for comparison and consistency throughout the text ANOVA test was also performed on datasets of parameterisation given in Eq. (2) and (3).

The results of ANOVA test on these datasets are now discussed in detail in the supplementary text under section 4 between lines L96 – 109. In the manuscript this section is mentioned on lines 248 to 255.

*2) Similar statistical analysis is also needed for the different analyses north and south of the polar front. Pearson coefficients for the correlation of observed IO with various parameters divided and combined data set are shown in Fig. 8. These help give some idea of the differences between the correlations on either side of the front, but the picture is incomplete.*
*Two particular results highlighted in the text are demonstrative: GEOS-Chem modeled IO is significantly correlated with observed IO in the data subsets but not in the complete data set. Fig. 8 shows this also the case for CAM-Chem at ~94% confidence as well. The reason for this, as stated by the authors, is that the variability in both models across the polar front is significantly different than observed. Further, the reader can see this for themselves in Figs. 4 and 10. In contrast, correlations with chl-a are much more difficult to interpret. Most data occupy a limited dynamic range in Fig. 4 and the correlation plots in Fig. 9 indicate individual points may be driving the correlation but this is not clear. In both instances systematic statistical assessment of the data subsets would be helpful.*

RESPONSE: The reviewer is right to point that individual data points of chl-*a* are driving the correlation with observed iodine oxide (IO) levels. This point is mentioned in the manuscript on lines 643 – 644. We mentioned that high IO levels were observed in a narrow band where elevated chl-*a* observations were noted close to the Kerguelen Islands at 43° S. Here, the figure was incorrectly marked as Fig. 5 instead of Fig. 4e and this error is now corrected in the manuscript. We agree that systematic assessment of individual regions, and sub-regions would be useful but unfortunately this is not possible due to the low number of datapoints (as pointed out by the reviewer in the first comment).

*In addition, I have the following additional two major comments.*

*3) As the authors state in the abstract their results start from "the first concomitant observations of iodine oxide (IO), O3 in the gas phase, and sea surface iodide concentrations." The choice of "concomitant" implies some intrinsic link between the set of parameters, and theoretically these parameters are expected to correlate via the mechanism outlined by Eqs. 7 and 8. However, the authors ultimately find that the correlations are the opposite of those in the equations as shown in Fig. 8 and discussed in the text. Keeping close to the underlying data these correlations should be shown in full similar to Chl-a in Fig. 9.*

RESPONSE: The word 'concomitant' was used to highlight the previously established links between sea surface iodide and surface ozone leading to the flux of HOI and $I_2$ (as evident in Eq. (7) and (8) from literature). This does suggest an intrinsic link between the set of parameters. We agree that the findings in this study are contrary to these expectations and so a correlation of sea surface iodide and ozone concentration with flux of HOI and $I_2$ (Fig. 7) is included in the manuscript to highlight this point. However, we do not feel that all the correlations which are not significant need to be presented as separate figures as in Fig. 9 as in the case of chl-*a*. All the significant correlations are presented as separate figures and figure 8 shows the parameters which are not significant in a single plot showing the summary.

*4) The underlying measurements are contained in four other papers (Chance et al., 2019a,b and Mahajan et al., 2019 a,b) are fundamental to assessing these findings. These are*

*sufficiently critical to the results presented that I would recommend linking them as companion papers. I cannot locate Chance et al. (2019b) and do not believe it is published. The measurements are described in part in Chance et al. (2019a) nonetheless it is troubling that such critical measurements are neither published nor described more fully. If the measurements are not yet published the description in this paper should be expanded. Notably, MAX-DOAS data (which are published elsewhere) appear in the supplement while the SSI data do not.*

RESPONSE: A preprint of the Chance et al. 2019b paper is now available on ESSOAr. The manuscript is currently under review in Frontiers of Marine Science and is cited as Chance et al., 2020 in the manuscript, first appearing on line 128. The sea surface iodide (SSI) data are not included in detail in this manuscript as the Chance 2019 and 2020 papers (cited in the manuscript) have a full description of the dataset – these are both available now.

*I have following minor and technical comments.*

*Line 28: As commented above, "concomitant" here may be misleading in the abstract. While the latter portion of the abstract (L 37-39) makes clear that "Sea-air fluxes ... calculated from the atmospheric ozone and seawater iodide ... failed to adequately explain the detected IO in this region" it does not make clear that the correlations are largely null or even contrary to expectations.*

RESPONSE: We are not sure what the reviewer means since the observations of iodide, ozone and IO were indeed made at the same place and time, hence the use of the word concomitant.

*Line 58-60: Saiz-Lopez et al. (2006b) included IO condensation onto particles in order to explain particle growth. However, I do not believe there is any claim of direct nucleation by IO. This sentence should be clarified.*

RESPONSE: The reviewer is mistaken in this claim – $I_2O_2$ was considered as 'the condensable unit in the iodine particle nucleation' in the model used in that publication.

*Line 73 and 74: Why are these reactions not labeled and numbered?*

RESPONSE: These reactions are a part of reaction R1 that show the steps leading to formation of HOI. This is now added and reactions are labelled in the text.

*Line 266: The detection limit and precision should match units to be more easily compared.*

RESPONSE: Thank you for bringing this to our attention. The units are changed to ppbv for both detection limit and the precision on line number L279.

*Line 270-272: Consider simplifying description of wind flagging to inclusion of the forward hemisphere or the like. Presently mildly confusing.*

RESPONSE: Added line 284.

*Line 302-30: Wind speed is first introduced here but discussed frequently hereafter. What is the wind speed referred to? e.g. is it U10 or some other standard? This is particularly relevant later when comparing with model treatment of wind parameters.*

RESPONSE: The wind speed data is referred to the winds arriving at the ship's AWS sensor located at the height of approximately 10 m above the sea. Hence this data is U10 data as per the standard treatment. This information is now included in the manuscript on line number 404.

*Line 305 - 307: As is made clear later on line 317, neither of the activation energies determined in MacDonald et al. (2014) is significantly different from zero. While it does not examine the products separately, Magi et al. (1997) does find a significant activation energy for the first I- + O3 reaction. MacDonald et al. noted that the Arrhenius pre-exponential factor in Magi et al. is ten orders of magnitude greater than the diffusion limit to justify assuming an activation energy for the first step of zero. More recent work has also called the rates determined in Magi et al. into question (Moreno et al., 2018), however, the confounding factors (high iodide concentrations and iodide surface activity) cannot fundamentally dispel the observed positive temperature dependence. Notably, the values reported in MacDonald are predicated on an assumption that the activation energy of I- + O3 is zero and it is even more uncertain whether the overall activation energy to produce I2 is negative. The activation energies from MacDonald are better summarized as approximately zero (e.g. Moreno and Baeza-Romero 2019) as the overall temperature dependence remains unresolved.*

RESPONSE: We thank the reviewer for pointing out this paper. A short description on the above is now added to the manuscript (Line 333).

*Line 401-405: While diurnal variation in O3 can be inferred, its reversal is not apparent in Fig. 5b as referred to.*

RESPONSE: We agree with the reviewer that the lack of diurnal variation is not clear in Fig. 5b. An inset is included in this plot to highlight the same and figure is revised.

*Line 645 - 646: How do the authors infer that photochemistry is responsible for the differences? This is not obvious to me.*

RESPONSE: This has now been changed to 'that either photochemistry or dynamical dilution of the fluxes'.

*Line 654 – 655: From Fig. 7 it appears that the p-value for the HOI correlation is 0.04. Given that 5% significance is used as a standard elsewhere in the paper this would indicate that HOI does show significant correlation contrary to this statement.*

RESPONSE: Figure 7 shows the correlation of HOI against I⁻ and not of HOI against IO.

*Line 771: VOI not previously introduced.*

RESPONSE: This has been edited in the latest version of the manuscript.

*Figure 3: The literature calculations are not readily compared with the observations as they are in separate panels. The observations should appear in both panels.*

RESPONSE: This has been changed accordingly.

*Figure 9: It should be made clear that the IO here is observed as modeled IO is also presented elsewhere. The legends readily blend in with the scattered data and should be made more clearly separate.*

RESPONSE: This has now been made clear in the caption.

*Table 2: The database location column should be moved left of the equations as it is otherwise unclear in isolation what the differences between Eq. 2, 3, and 5 are. Iodide and nitrate concentrations should be given with consistent (though not necessarily the same) units, i.e. M or mol L-1 but not both.*

RESPONSE: We have changed the table as requested. The units of the iodide and nitrate concentrations were chosen to be consistent with the equations from past publications which estimate the iodide using different parameters but have clarified this in the caption.

*Table 2 and Supplement: In addition to the expectation of higher R2 values for fitting to the same data set, equations having more degrees of freedom are expected to have better fits. The adjusted R2 values should be used for proper comparison of the overall parameterizations. If this is already the case it should be described as such in the supplement.*

RESPONSE: Corrected.

*Reviewer #2:*

*The paper by Swaleha Inamdar et al shows new and simultaneous measurements of iodine oxide (IO), ozone (O3) in the gas phase, and sea surface iodide (I-; SSI) concentrations during the ISOE-9 ship campaign in the Indian Ocean and Southern Ocean in January-February 2017. These measurements are complemented with previously published ship based measurements in the Indian Ocean and Bay of Bengal and with different available parametrizations to compute the iodine (I2) and hypoiodous acid (HOI) fluxes. This study includes important new results which should be publishable after a detailed and careful major revision of the manuscript taking all comments into account.*

RESPONSE: We thank the reviewer for the comments and have tried to address the comments below.

*General comments:*
*Earlier studies: The paper misses to refer to other iodine ship-based studies, such as Hepach et al (2016), where iodocarbons, IO, and many different biological parameters were observed and possible biological production mechanisms were discussed. A positive correlation between iodine sources and biology and a biology control is not a new result. This has to be taking into account in the abstract, introduction, discussion, and conclusions of your results. There were earlier ship based measurements of atmospheric IO, I2 and ozone, f.e., in the tropical West Pacific during e.g. the SHIVA campaign (Pfeilsticker et al 2013) and in the Indian Ocean with the OASIS campaign (Krüger et al 2015) which should be mentioned and related to.*

RESPONSE: We thank the reviewer for pointing out some of the papers that need to be cited and have included them. The Pfeilsticker et al 2013 and Krüger et al 2015 citations are overviews and do not present any IO data.

*Indian Ocean: What about the strong seasonality of the Indian Ocean, physical and biological, which may play an important role for the interpretation of your results? This needs to be included in the introduction and the discussion (see Schott and McCreary 2001; SIBER Report No. 1, 2011). Your paper should go beyond a correlation based only discussion. What are the mechanisms in the Indian Ocean: Any biology, ocean and atmospheric circulation impacts? It would be very interesting to get some more details on the spatial distribution of your observed in-situ quantities compared to satellite and/or global model data, adding maps of e.g. SST, Chl-a, wind, SSI/I2.*

RESPONSE: We agree that the seasonality of the Indian Ocean needs to be studied and that these data would be useful to compare with model data and maps of etc. We have already used results from the global models CAM-Chem and use the SST and chl-a from satellites in the discussion. However, we have observations from only during the December-March period and hence cannot speculate more on the seasonality. This is however a study that needs to be done in the future.

*Measurement, flux parametrization, and model details, errors and uncertainties: What are the error bars of the measurements especially of SSI and what are the uncertainties of the flux estimates? This needs a careful and detailed discussion in the ms. The observed SSI (Chance et al 2019b under review) study is not available to the readership so that we cannot*

*find any information about the kind of measurements nor the quality. What was the measurement strategy (day vs night time, how often etc)?*

RESPONSE: Full details of the iodide measurements are described in the companion paper Chance et al, 2019b (changed to Chance et al., 2020 in the latest version) that is now available as a preprint on ESSOAr and the manuscript is currently under review in Frontiers of Marine Science. The uncertainties of the iodide method were estimated by repeating each scan 5-6 times, with scan repeatability equal or better than 5%. Calibration was by 2 or 3 standard additions of a KI solution (~10-5or 10-6M). The errors reported here reflect the standard deviation of the repeat scans and the standard error on the intercept and slope of the calibration. Precision was estimated by repeat analysis (n = 6) of selected seawater samples over period of ten days and was found to be lower than 7% relative standard deviation. The following sentence has been added in te manuscript (L189): "*The errors reported here reflect the standard deviation of the repeat scans and the standard error on the intercept and slope of the calibration.*" Most samples were diurnal, except for some taken during two time series on the SOE-9 cruise (n=11). The uncertainties on the calculated fluxes have not been estimated here, for two reasons (1) the methods used to calculate fluxes would give very incomparable error types (machine learning vs multiple regression) and (2) the multiple regression proposed in Carpenter et al., 2013 to calculate the iodine fluxes (Eq. 7  and 8) does not mention the associated errors.

*Where were the surface water iodide measurements carried out onboard of the ship and when? Does the diurnal cycle play a role? Substantial measurement details are missing and need to be added to understand your ship measurement and study design better.*

RESPONSE: Full details of the iodide measurements are described in the companion paper Chance et al, 2019b (changed to Chance et al., 2020 in the latest version) that is now available as a preprint on ESSOAr and manuscript is currently under review in Frontiers of Marine Science. Briefly, surface water samples were obtained manually from the upper 30-70 cm of the sea surface using a metal bucket deployed over the side of the ship upwind near the stern, during the SOE-9 and BoBBLE cruise. Additionally, samples were obtained during using the first depth of CTD rosette casts (estimated at 20m) at 17 (SOE-9) and 8 (SK-333) CTD stations. Manual surface samples were taken (by bucket) at least twice a day along the entire cruise track (except when the ship was stationary for CTD stations). Sampling included two time-series, one at ~40oS, and one in coastal Antarctic waters at ~68oS (around the Polar Front), during which samples were collected at 4 or 6 hour intervals for up to 72 hours. No clear diurnal trends could be discerned, in accordance with previous studies (e.g. Brandão, Ana Claudia M., Angela de Luca Rebello Wagener, and Klaus Wagener, 'Model Experiments on the Diurnal Cycling of Iodine in Seawater', Marine Chemistry, 46.1–2 (1994))

*For the observed meteorological data, surface wind is conventionally given as 10 min averages and then there are gusts (instantaneous wind). Currently, you use hourly averages which lead to a smoothing of the average wind speed and thus impact your flux parametrization calculations which are based on a threshold limit of 14 m/s. Next, at which altitude levels onboard of the ship were your wind and others quantities measured? Conventionally 10 m surface wind is used for flux calculations. What did you use and on what are the flux parameterisations based on? The measurement section, data and graphs need a thorough and detailed revision.*

RESPONSE: We agree that the wind speeds were averaged as we use hourly averages to calculate the fluxes. However, this is necessary due to the frequency of the other observations.

However, the original data is measured at a high frequency and the winds were measured at U10, which is now mentioned in the manuscript.

*Substantial details are also missing for the flux parametrization. How well do the estimated iodine fluxes explain observed surface atmospheric I2 concentrations? What are the largest uncertainties also in contrast to the common bulk parametrizations of air-sea fluxes which have a very high (>50%) uncertainty especially with regard to the role of the wind?*

RESPONSE: $I_2$ was not observed during the cruise as mentioned in the manuscript. A detailed discussion in the flux estimation equations is already presented in (Carpenter et al., 2013) and (MacDonald et al., 2014) and we have made use of those equations to study their applicability to the Indian Ocean, and hence have not expanded beyond the discussion in the methodology section which presents the largest sources of uncertainties. Furthermore, both parametrizations do not give the uncertainties associated with the specific parameters.

*What are the main chemistry transport and chemistry climate model uncertainties? What is the role of the meteorology and ocean surface (composition and circulation); is this consistently taken into account in these two models compared to your observations?*

RESPONSE: We agree that all models have uncertainties resulting from transport and chemical reactions used. However, a detailed analysis of model uncertainties is beyond the scope of this paper. Model description papers have been cited in the manuscript.

*Specific comments:*
*Line 127-129: Grossmann et al 2013 and others published remote open ocean data. Please rephrase the sentence.*

RESPONSE: Changed

*Table 1: There are no 2014 measurements listed in the third column although you mention this in the table caption, abstract, introduction etc.*

RESPONSE: Corrected

*Technical corrections:*
*Figures and figure captions: The graphs and figure captions are not self-explaining and not presented in a consistent way. The acronyms are mostly not introduced nor are the figures easy to relate to each other, f.e. ozone in Figure 4 and 5 is it the same? What is PF; I assume Polar Front and where is this in Fig 5? All your figures and figure captions need a thorough revision.*

RESPONSE: Yes, the data in the figures is the same. The polar front is marked only in the figures which have oceanographic data, as this front has no atmospheric significance for the measured parameters. As suggested by the reviewer, we have checked through the captions and made changes where we felt the details were not sufficient.

*Reviewer #3*

*The paper by S. Inamdar is using a large data set of seawater iodide, atmospheric ozone and atmospheric IO concentrations to test the reactive inorganic iodine fluxes calculated from different parameterisations of seawater iodide,. The authors propose new parameterisations of seawater iodide that are specific for given regions of the global ocean, and compared to already established parameterisation for the global ocean. They find that the parameterisation used has little impact on the computed atmospheric IO concentrations. Observed IO concentrations cannot be adequately computed using inorganic iodine fluxes and chemistry. As IO is correlated to Chl-a, the authors suggest a biogenic impact on iodine in the region investigated. The paper is well and clearly written and organized. Iodine fluxes, chemistry and impacts on the atmospheric composition are poorly understood and this study brings a nice input into our understanding. I suggest the paper is published after only minor comments (below) are taken into account.*

RESPONSE: We thank the reviewer for the positive comments and have answered the specific comments below and made the corresponding changes in the manuscript.

*Minor comments*

*Section 2.1 iodide parameterisations*

*Lines 201 to 218 : the argumentation on the need to have regional parameterizations should go in the introduction ?*

RESPONSE: Changed.

*Line 226 : would be nice to recall why sea surface nitrate concentrations were chosen as a parameter influencing iodide concentrations*

RESPONSE: Added.

*Section 2.2 ozone measurements*
*Contaminations on a ship may occur from other sources than the ship's smokestack (such as cooking exhausts, or air conditioning exhausts). Were there any indicator of anthropogenic compounds concentrations available to exclude contaminations?*

RESPONSE: Yes, ozone shows very strong effects of the smokestack and these were removed from the observations during data cleaning as mentioned in the manuscript. The cleaning was done using the quick titration of the $O_3$, which was visually easy to identify, the black carbon observations and the aerosol number observations.

*3.Results*
*3.2 Iodide line 432-433: the end of the sentence is not clear, please reformulate*

RESPONSE: Corrected.

*3.3 Iodine fluxes line 491: premature to mention discrepancies between modelled and measured IO in this section? Would better fit in the discussion section*

RESPONSE: Corrected.

*4. Discussion*
*line 712: concerning the lack of correlation with satellite base Chl-a while in situ Chla oncentrations are correlated to observed IO concentrations. May this be due to geographical differences in what biological species Chl-a represent in these different regions, or may be due to uncertainties in the Chl-a retrieval from satellite, or even also scaling problems. Did the authors try to extract satellite Chl-a where the actual Chl-a in situ measurements were performed to compare one with the other?*

RESPONSE: This is correct and the sources of the difference between the satellite and in situ could be many as the reviewer has suggested. The chl-a data from the satellites was extracted from the same location as the in situ observations. This is now added in the manuscript.

[revised manuscript text omitted]
.  In this case,  latitude and salinity were the only parameters that showed significant dependence on  the observed SSI (Table S1). Individual parameters with significant $R^2$ values were used to obtain a parametric equation for SSI concentration. The first second, and third scenario resulting in parametrisation denoted by Eq. (2) Eq. (3), and Eq. (3a) respectively are given in the main text in Table 2.  A combination similar to the Chance parameterisation given in Eq. (2) gave maximum $R^2$ value of 0.794 (N = 128) for the Indian Ocean and the Southern Ocean region. In this equation, all parameters are significant except for salinity and nitrate concentration. Removal of any one of these insignificant parameters did not make the other significant. The coefficient for this equation (Eq. 2) also remained insignificant with high error value (22 ± 137). The combination of $SST^2$, latitude, nitrate and salinity resulted in a maximum $R^2 = 0.86$ (N=110) for the dependent variable [iodide] in Eq. (3). The inclusion of $MLD_{pt}$ (with highest $R^2$ for MLD) increased the $R^2$ slightly but had a non-uniform distribution of the residuals and was thus excluded. Similarly, the addition of chl-*a* to the equation did not change the $R^2$ significantly, and thus chl-*a* was removed from the final equation. The Indian Ocean scenario parameterisation in Eq. (3a) obtained $R^2 = 0.325$ (N=18). All parameters (latitude, salinity) and the coefficient were insignificant with large error values as shown in Table 2. These equation datasets were tested for statistical robustness by ANOVA test using StatPlus analysis software. Both equation (2) and (3) dataset result in higher F ratio value corresponding to the critical F value from f-distribution table and p-value < 0.0001 at 0.05 significance level. Eq. (2) obtains F = 94 with (5, 122) degree of freedom (critical value = 2.289) and Eq. (3) obtains F = 161 with (4,105) degree of freedom (critical value = 2.458). However, Eq. (3a) Indian Ocean dataset provides statistically insignificant result as the F value 3.604 with (2,15) degree of freedom is lower than the critical value of 3.682 with p = 0.053. Thus, this parameterisation is omitted from further analysis in the study and is indicative that the sea surface iodide estimation in the Indian Ocean does not follow the Chance parameterisation technique. It is important to note that this analysis involved a small dataset (N=18) and more observational studies will be required to estimate iodide concentrations in this region. SSI concentration was also estimated using the logarithmic parameterisation by Chance et al. (2014) and it was found to be  higher in comparison to the measured SSI concentration from ISOE-9. The ln[iodide] equation estimated SSI concentrations of ~500 nM in the Indian Ocean region which is very high compared to global observations of SSI in the Indian Ocean (Chance et al., 2014, Chance et al., 2019) and in comparison to the observations from SK-333 and BoBBLE for the South Indian Ocean. Therefore, we excluded the logarithmic parametrization for this study and  suggest that  the ln[iodide] parametrization is not adequate for SSI estimation.

[Figure]

**Figure S1: Map of the south Indian Ocean and the Southern Ocean showing the cruise track (black line) for the ISOE-9 campaign. Along the cruise track 5-days backward wind trajectories (HYSPLIT) of the air masses arriving the locations at noon each day of the ISOE-9 expedition. Sea surface iodide sampling locations marked in red circles along with the date of sampling.**

[Figure]

**Figure S2: Timeline of the O₄ and IO DSCDs observed during the ISOE-9 expedition.**
**The top scale indicates corresponding latitudes for the dates, and colour code represents**
**the elevation angle (°) for each scan. Smaller circles indicate DSCDs below σ detection**
**limit for IO and 2σ in case of O₄; bigger circles indicate DSCDs above the detection**
**limit respectively.**

[Figure]

**Figure S3: An example of typical spectral fit for O₄ (a) and IO (b) during the ISOE-9**

**expedition. These spectral fits were taken on 26 February 2017 at 15:35 (local time), for**

**solar zenith angle 69.5º and 1º elevation angle. These fits retrieved O₄ slant column**

**density of $(4.35\pm0.035)\times10^{43}$ molecules $cm^{-2}$ and $(2.24\pm0.36)\times10^{13}$ molecules $cm^{-2}$ with**

**residual optical density (root mean square) of $3.2\times10^{-4}$ and $5.5\times10^{-4}$ respectively.**

 **7. Tables**

| Parameter | $R^2$ | Slope (m) | Intercept (C) | p < 5%? (p) |
|---|---|---|---|---|
| **SST** | 0.64 | $4.26 \pm 0.29$ | $31 \pm 4.77$ | Yes (0) |
| | *0.62* | *$4.03 \pm 0.304$* | *$32.2 \pm 4.17$* | *Yes (0)* |
| | 0.07 | $28.8 \pm 26.22$ | $-668.85 \pm 754.52$ | No (0.29) |
| **1/SST (K$^{-1}$)** | 0.62 | $-345781 \pm 23910$ | $1297 \pm 83.9$ | Yes (0) |
| | *0.59* | *$-322918 \pm 25302$* | *$1215 \pm 89.5$* | *Yes (0)* |
| | 0.07 | $-2616459 \pm 2392959$ | $8826 \pm 7926$ | No (0.29) |
| **SST$^2$** | 0.73 | $0.16 \pm 0.0085$ | $41.1 \pm 3.6$ | Yes (0) |
| | *0.79* | *$0.18 \pm 0.01$* | *$39.2 \pm 2.7$* | *Yes (0)* |
| | 0.07 | $0.51 \pm 0.45$ | $-261.9 \pm 375.2$ | No (0.28) |
| **NO$_3$** | 0.42 | $-3.24 \pm 0.34$ | $125 \pm 5.7$ | Yes (0) |
| | *0.39* | *$-2.63 \pm 0.32$* | *$110.6 \pm 5.8$* | *Yes ($3.06 \times 10^{-13}$)* |
| | 0.03 | $19.34 \pm 27$ | $153 \pm 17$ | No (0.48) |
| **|Latitude|** | 0.55 | $-2.1 \pm 0.17$ | $178.3 \pm 8.3$ | Yes (0) |
| | *0.52* | *$-2.43 \pm 0.22$* | *$196.1 \pm 11.7$* | *Yes (0)* |
| | 0.30 | $8.74 \pm 3.35$ | $108.5 \pm 23.11$ | Yes (0.02) |
| **Monthly MLD$_{pt}$** | 0.17 | $-1.1 \pm 0.22$ | $125 \pm 9.2$ | Yes ($1.2 \times 10^{-6}$) |
| | *0.08* | *$-0.63 \pm 0.21$* | *$97.6 \pm 9.4$* | *Yes (0.003)* |
| | 0.14 | $-2.69 \pm 1.68$ | $203.41 \pm 30.38$ | No (0.13) |
| **Monthly MLD$_{vd}$** | 0.04 | $-0.48 \pm 0.2$ | $98 \pm 8$ | Yes (0.03) |
| | *0.003* | *$-0.11 \pm 0.19$* | *$75.9 \pm 7.5$* | *No (0.56)* |
| | 0.16 | $-2.69 \pm 1.55$ | $193.52 \pm 23.6$ | No (0.10) |
| **Monthly MLD$_{pd}$** | 0.12 | $-0.67 \pm 0.16$ | $110 \pm 7.8$ | Yes ($5.2 \times 10^{-5}$) |
| | *0.05* | *$-0.35 \pm 0.15$* | *$87.1 \pm 7.7$* | *Yes (0.02)* |
| | 0.15 | $-2.51 \pm 1.52$ | $194.8 \pm 25$ | No (0.12) |
| **Salinity** | 0.08 | $16 \pm 4.8$ | $-468 \pm 165$ | Yes (0.001) |
| | *0.23* | *$21.8 \pm 3.8$* | *$-675 \pm 130$* | *Yes ($8 \times 10^{-8}$)* |
| | 0.30 | $-42.41 \pm 16.21$ | $1609.3 \pm 551$ | Yes (0.02) |

| Chlorophyll -a | 0.025 | $-37 \pm 26$ | $84 \pm 8.6$ | No (0.16) |
|---|---|---|---|---|
| | *0.002* | *$-7 \pm 20$* | *$62 \pm 7$* | *No (0.73)* |
| | 0.01 | $77.83 \pm 206$ | $136 \pm 31$ | No (0.71) |

**Table S1: Linear regression analysis results for each parameter against field observations of sea surface iodide for paramterisation Eq. (2) in standard font and Eq. (3) in italics, and grey shaded rows for Eq. (3a). $R^2$ represents the coefficient of determination (COD); the last column is a check for statistical significance at 5% with the p-value in parenthesis.**

---

## Author Response (AR2)

**Response to reviewer comments for manuscript number: acp-2019-1052**

Comments by the reviewers are shown in an italic typeface and the responses are shown in a normal typeface.

We thank both the reviewers for providing further constructive comments and suggestions to our modified manuscript and are glad that they both find it worthy of publication pending minor changes. The following is a point by point response, with corresponding changes made to the manuscript. We hope that the manuscript will now be acceptable with these changes.

*Report #1*

*The authors should address the following general points before publication can be suggested:*

We thank the referee for the comments and have responded to them below.

*-Measurement, flux parametrization, and model details, errors and uncertainties:*

*>Discuss the impact of using 1 hourly averaged winds. Add the measurements details (shortly) into your paper as well.*

Response: We referee is right in suggesting that using hourly averaged winds for flux parameterisation would add to the uncertainty. The main issue is that the high temporal resolution variability of the instantaneous flux would be underestimated. The reason for doing this is to match the temporal resolution to the model output. We have now added a sentence regarding this caveat in the manuscript. 'The fluxes were calculated using the hourly wind speeds for the results to be comparable with model outputs as described below. This would result in a loss of high temporal resolution emission variability, but considering the frequency of the iodide and IO observations, computing the fluxes at a higher resolution would not give any extra information.' (line 505-508)

*-Indian Ocean:*

*>Discuss the impact of Indian Ocean variability (seasons and regions) of currents and Chla on their results (see e.g. SIBER report for details).*

Response: Whilst exploring the variability in the Indian Ocean on a regional and seasonal scale is important, we do not have enough observations to discuss these in detail. All our observations are during the same season and hence any such discussion would be speculative in nature. The cruises discussed in the paper are held during the austral summer and show similar chlorophyll-*a* concentrations (Figure 4). Hence, we feel that this discussion is beyond the scope of the paper, and something that should be explored in the future.

*-Earlier studies:*

*>The mentioned IO, I2 data are "published" in the corresponding ship cruise reports.*

The $I_2$ data is not measured but estimated. This paper brings together past reports along with new data from the ISOE-9, SK-333 and BoBBLE cruises. IO data for cruise ISOE-9 are not reported before and measurements of iodide were done for the first time in the Indian Ocean. The earlier reports of IO have been cited in the manuscript.

**Report #2**

*My previous recommendations for major revisions have been substantially addressed. The publication of Chance et al., 2020 in the interim has obviated my remaining concern. Improvements to the figures and the transparent presentation of statistical method in particular are to be praised. I have the following minor technical comments.*

We thank the reviewer for the kind comments and are glad to see that the changes made have made the manuscript more understandable. The minor corrections suggested have been made according to the points below.

*Line 635: While likely clear from context, it may help to specify here that these are surface wind speeds in the models. Are these wind speeds meant to be fully comparable to U10? Clarification of this here or elsewhere would be helpful.*

Response: Yes, the winds used in models are surface winds and are the closest match that the models have to U10. This has been now made clear on line 635. The line added is '…and hence fluxes calculated using the surface winds in these models are expected to be slightly different.'

*Line 1136: "observed" here is in red when it should be in black.*

Response: Changed.

*Table 2: 'sst' here should be consistently capitalized as 'SST' in the table, caption, and footnotes for consistency with the rest of the manuscript.*

Response: Changed.

References

Chance, R., Liselotte, T., Sarkar, A., Sinha, A. K., Mahajan, A. S., Chacko, R., Sabu, P., Roy, R., Jickells, T. D., Stevens, D., Wadley, M. and Carpenter, L. J.: Surface Inorganic Iodine Speciation in the Indian and Southern Oceans from 12o N to 70o S, Earth Sp. Sci. Open Arch., 36, doi:10.1002/essoar.10502894.1, 2020.

Mahajan, A. S., Tinel, L., Hulswar, S., Cuevas, C. A., Wang, S., Ghude, S., Naik, R. K., Mishra, R. K., Sabu, P., Sarkar, A., Anilkumar, N. and Saiz Lopez, A.: Observations of iodine oxide in the Indian Ocean Marine Boundary Layer: a transect from the tropics to the high latitudes, Atmos. Environ. X, 1(January), 100016, doi:10.1016/j.aeaoa.2019.100016, 2019a.

Mahajan, A. S., Tinel, L., Sarkar, A., Chance, R., Carpenter, L. J., Hulswar, S., Mali, P., Prakash, S. and Vinayachandran, P. N.: Understanding Iodine Chemistry Over the Northern and Equatorial Indian Ocean, J. Geophys. Res. Atmos., (x), 2018JD029063, doi:10.1029/2018JD029063, 2019b.

---

## Author Response (AR3)

**Response to reviewer comments for manuscript number: acp-2019-1052**

Dear Dr. Engel,

Thank you for the final comments and for accepting the manuscript. We have now added a brief about the wind measurements and made the second change as you have suggested below

Best wishes,

Anoop
* * *
there are two small remaining issues:

1.: Rev. #1 suggested adding some details on the measurements. Please consider this suggestion.

A brief on the wind measurements have been added as:

During the three expeditions, meteorological parameters of ocean and atmosphere were measured using an on-board automatic weather station (WeatherPak®-2000 v3), which is specially built for shipboard observations and manual observation techniques. The WeatherPak system was installed in the front of the ship, with the sensors approximately 10 m from the sea surface. The weather system is equipped with a GPS system for measuring the true wind speed and direction along with the apparent data. The SST and salinity were measured manually through bucket sampling.

2.: In the short paragraph added to discuss the use of 1 hourly averaged winds, I believe that the "would" in the second sentence should be replaced by "will".

Changed.

best regards,

Andreas Engel